# Technology Identification, Evaluation, Selection, and Optimization of a HALE Solar Aircraft

Ju-Yeol Yun and Ho-Yon Hwang * 

Department of Aerospace Engineering, Sejong University, 209, Neungdong-Ro, Gwangjin-Gu, Seoul 05006, Korea; yunjy0517@sju.ac.kr
* Correspondence: hyhwang@sejong.edu; Tel.: +82-2-3408-3773

**Abstract:** In this paper, sensitivity analysis and optimization of a high altitude long endurance (HALE) solar aircraft was implemented. Zephyr S was referred to for the aircraft conference configuration, and OpenVSP and XFLR5 were employed to create configuration and perform aerodynamic analysis. In the conceptual design stage of the HALE solar aircraft, technology identification, evaluation, and selection (TIES) methodology was employed. According to the design requirements, problem definition was established, and design goal, variations, and targeted values were set up to implement independent design variables to meet the design requirements. Based on the design of experiments (DOE), modeling of the relationship between design objective parameters and independent design values was implemented. The independent design variables with the largest influence were selected in the screening test. By employing the selected independent design variables, regression equations and sensitivity profiles were produced through response surface method. Inter-factor relationship was easily analyzed through the sensitivity profile. Regression equations were employed in the Monte Carlo simulation to draw design objective parameter values for 10,000 combinations of independent design variables. As a result of the Monte Carlo simulation, the design feasibility of design objective parameters was assessed. Optimization was performed using the desirability function of JMP software, and constraints were applied to each design objective parameter to derive the optimum values of independent design variables. Then, the values of optimized design independent variables were applied to the solar aircraft design framework and analyzed for the endurance flight performance. By comparing the endurance of the optimized configuration with the reference configuration, it was confirmed that the endurance could be improved by using the methodology proposed in this study.

**Keywords:** solar aircraft; aircraft design; solar cell; design of experiments; TIES method

---

## 1. Introduction

The high altitude long endurance solar aircraft (hereinafter, HALE solar aircraft) is a long endurance aircraft with a power source of solar energy, flying at 18 km or higher altitude. Solar aircraft have attracted attention over the past several decades because of their promising potential in military and civilian applications. The most appealing feature of solar energy is that it is continuously available during flight, and thus may yield a nearly fuel-less emissions-free flight [1]. In recent years, with the rapid improvement of environmental protection and long endurance following the "Pathfinder" [2] and "Helios" [3] models, the world saw research upsurge in the area of solar aircraft. This type of aircraft utilizes electric energy transformed from solar radiation via photovoltaic (PV) cells and stores it in secondary batteries during daytime flight, has better endurance than conventional aircraft, and may even be able to fly permanently [4]. The long endurance capability of the solar aircraft platform has been envisioned as a possible alternative to communication satellites. They could also monitor weather, track hurricanes, and make substantial contributions in disaster management via more precisely

directing emergency resources [5]. In comparison to orbital satellites, "atmospheric satellites" would offer better observational resolution, local persistence, and the capability of reuse [6].

The first flight of a solar aircraft was Sunrise I. After that, an improved version, Sunrise II, was built and tested on the 12th of September 1975. On the 18th of May 1980, the Gossamer Penguin realized which can be considered as the world's first piloted, solar-powered flight. On 7 July 1981, the next version, named Solar Challenger, crossed the English Channel with solar energy as its sole power source. In 1993, the Pathfinder, with its 30-m wingspan and weight of 254 kg, was tested at low altitude [7].

In 2004, the Autonomous Systems Lab of EPFL/ETHZ launched the Sky-Sailor project under a contract with the European Space Agency. This project presents the methodology used for the global design of solar powered airplanes to achieve continuous flight on earth [7].

QinetiQ is also very active in the field of solar HALE platforms with Zephyr, an airplane that flew in July 2006 for 18 h. Zephyr was acquired and developed by Airbus and Zephyr S is regarded as the world's most advanced aircraft in terms of long endurance flight performance. In the test flight in 2018, Zephyr S set an endurance record of 25 days, 23 h and 57 min; however, the location of test flight was Wyndham in Australia, near the equator. If flown on a mission at a different latitude than the equator, it seems the aircraft would show a huge difference in its endurance performance for the different flight conditions. Thus, the location and date of the flight determine the aircraft's performance. The available power is dependent not only on the factors of wing area (solar cell area), cell efficiency, latitude, and time of year but also on the time of day [8].

Another variable with a great effect on endurance is the aerodynamic characteristics of the aircraft. In the case of solar aircraft, most drag is generated by the main wings; this can be modeled with a simple aerodynamic analysis program such as XFLR5. However, inconveniently, calculation must be repeated each time a variable changes. To resolve this problem, a new aerodynamic model is designed in consideration of a configuration design variable and flight condition variable; model is then utilized as the basic equation for long endurance performance analysis by assessing flight possibility and margin value [9].

As shown above, because of solar energy the power source of solar aircrafts largely varies depending on the operating environment factors such as flight date, location, and subsystem efficiency, all of which affect the endurance, it is necessary to perform sensitivity analysis of the main design variables. Given that it is the near-equatorial areas, with maximum solar energy incidence, where solar aircraft under development across the world are tested, solar aircraft sensitivity analysis were implemented in this study in consideration of flight mission latitudes other than those near the equator.

In addition, the mission profile is another important consideration. Climb rate, descent rate, and speed changes are calculated through the balance between the amount of available solar energy and the required energy. Studies on the descent are also necessary to minimize the energy required for cruise and night-time flight. In addition, changes in climb time are calculated as a function of takeoff start time and aircraft moving distances due to westerlies [10].

An optimization process is very important in the conceptual design phase of a solar aircraft because it can enable maximum aircraft performance [10]. The response surface methodology (RSM) was used as the optimization process [11]. Furthermore, in this study, the technology identification, evaluation, and selection (TIES) method was employed to analyze the solar aircraft conceptual design and sensitivity. The TIES method provides a comprehensive and robust methodology for decision-making in the conceptual aircraft design phase. This method provides the ability to easily assess and balance the effects of various technologies without need of complicated, time-consuming mathematical formulations [12].

The variety of design parameters must be considered during the conceptual design phase of HALE solar aircraft. Therefore, it is necessary to predict and analyze the sensitivity of sizing and design feasibility of a HALE solar aircraft for changes in various combinations of design variables. There are many studies analyzing the possibility of long endurance flight that do not consider those various detailed design variable combinations. So, this study introduced the design of experiment (DOE) to

obtain the maximum information from the minimum number of experiments. Using this statistical method, the design possibility of long endurance performance can be identified in advance, thereby the time and cost for the conceptual design of HALE solar aircraft can be saved.

The purpose of this study is to present a methodology that can derive optimal design-independent variables and improve aircraft performance using a statistical approach. In this study, the configuration for performance evaluation was referenced to Zephyr S, and configuration and aerodynamic properties were derived using OpenVSP [13] and XFLR5 [14]. For modeling and simulation, Microsoft's Excel, SAS's statistical program JMP [15], and in-house solar aircraft design framework software were used. Design independent variables that have a great influence on design target variables were derived through design of experiment and screening test, and design feasibility and sensitivity between variables were analyzed through response surface method and Monte Carlo simulation. Finally, the optimized design-independent variables were derived using the desirability function of JMP. Based on the optimized design-independent variables, the performance of the newly designed solar aircraft were analyzed and compared with the reference aircraft.

## 2. HALE Solar Aircraft Concepts

### 2.1. Reference Aircraft Configuration

HALE solar aircrafts must obtain solar energy necessary for long endurance and, at the same time, have sufficient wing area to create lift for flight. Most solar aircrafts have large wingspan appropriate to secure sufficient area to attach solar cells and gain lift. In this study, a reference configuration was established as in Figure 1 using OpenVSP [13], a program employed for configuration modeling, by referring to the configuration of Zephyr S. Table 1 shows detailed specifications of the aircraft configuration.

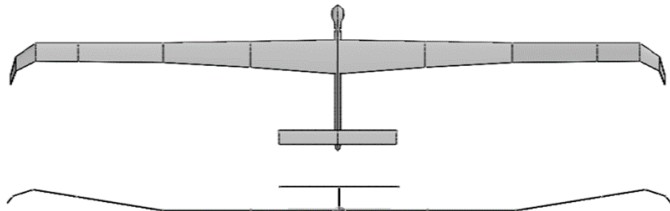

**Figure 1.** Reference high altitude long endurance (HALE) solar aircraft configuration.

**Table 1.** Reference HALE solar aircraft geometric data.

| Classification | | Value | Unit |
|---|---|---|---|
| Main wing | Wing span | 24.2 | m |
| | Chord length | 1.10 | m |
| | Area | 30.3 | m$^2$ |
| | Aspect ratio | 18.1 | – |
| | Airfoil | DAE-11 | – |
| | Taper ratio | 0.26 | – |
| Horizontal wing | Wing span | 4.20 | m |
| | Chord length | 0.80 | m |
| | Area | 3.36 | m$^2$ |
| | Aspect ratio | 5.25 | – |
| | Airfoil | NACA 0010 | – |
| | Taper ratio | 1.00 | – |
| Vertical wing | Wing span | 2.40 | m |
| | Chord length | 0.80 | m |
| | Area | 1.92 | m$^2$ |
| | Aspect ratio | 3.00 | – |
| | Airfoil | NACA 0010 | – |
| | Taper ratio | 1.00 | – |
| Fuselage length | | 8.15 | m |



### 2.2. Aerodynamic Analysis

HALE solar aircrafts that have been developed so far have average cruising speeds of 15~30 m/s and perform missions in low Reynolds number range. The DAEs airfoil has an excellent max lift coefficient, lift to drag ratio, and large thickness ratio in low Reynolds number range. Moreover, the HALE solar aircraft wing characteristic of being less sensitive to airfoil surface accuracy makes it easier to produce by applying solar panels. For these reasons, DAE-11 was selected as the airfoil of the main wing in this study. With respect to the DAE-11 airfoil, 2-dimensional aerodynamic data were produced using XFLR5 [14] and, based on these data, the lift and drag coefficients of a finite wing (3 dimensional) were calculated. The lift coefficient ($C_L$) can be calculated from Equation (2):

$$q = \frac{1}{2}\rho V^2 \tag{1}$$

$$C_L = \frac{W}{S}\frac{1}{q} \tag{2}$$

where $\rho$ is density, $V$ is flight speed, $W$ is the total weight of the aircraft, $S$ is the wing area, and $q$ is the dynamic pressure.

Drag coefficient ($C_D$), as in Equation (3), is divided into induced drag and parasite drag. The parasite drag is composed of form drag and skin friction drag [16]. Equation (3) can be expressed as a quadratic approximation of the lift coefficient ($C_L$) like Equation (4) [1]. In Equation (4), $K_1$ is the coefficient of drag-due-to lift and $K_2$ is assumed to be 15% of $K_1$.

$$C_D = C_{D,induced} + C_{D,form} + C_{D,skin} \tag{3}$$

$$C_D = K_1 C_L^2 + K_2 C_L + C_{D,skin} \tag{4}$$

where $K$ is the coefficient of drag-due-to lift and $K_1$ can be expressed as Equation (5) [1].

$$K_1 = \frac{1}{\pi ARe} \tag{5}$$

Induced drag ($C_{D,induced}$) increases in proportional to the square of the lift coefficient, as in Equation (6) [17].

$$C_{D,induced} = \frac{C_L^2}{\pi ARe} \tag{6}$$

where $e$ is the Oswald span coefficient and generally varies according to the wing aspect ratio [18].

$$e = \begin{cases} 0.9 & AR \leq 20 \\ 1.2 - 0.015AR & AR > 20 \end{cases} \tag{7}$$

Form drag ($C_{D,\,form}$) changes depending on the characteristics of the configuration, such as aircraft cross-section size. In this study, $K_2$ is assumed to be 15% of $K_1$; this is a typical value for a glider aircraft with large wings and small fuselage area [17].

$$K_2 = 0.15\frac{1}{\pi ARe} \tag{8}$$

$$C_{D,form} = 0.15\frac{C_L}{\pi ARe} \tag{9}$$

Skin friction drag ($C_{D,skin}$) is generated by skin friction between aircraft wing surface and air. Air viscosity is the cause. This is a critical factor of low-speed aircraft [19].

$$C_{D,skin} = \sum_{i=1}^{4} FF_i C_{f,i} \frac{S_{wet,i}}{S_{ref,i}} \tag{10}$$

where $i$ denotes fuselage, main wing, horizontal wing, and vertical wing respectively, $FF$ is form factor, $C_f$ is skin friction factor, $S_{wet}$ is wetted area, and $S_{ref}$ is reference area. $FF$, for Hoerner's streamlined body, was employed as in Equations (11) and (12) [20].

$$FF_{fuselage,i} = 1 + \frac{1.5}{(f)^{1.5}} + \frac{7}{(f)^3}(i = 1) \tag{11}$$

$$FF_{wing,i} = 1 + 2\left(\frac{t}{c}\right)_i + 60\left(\frac{t}{c}\right)_i^4 (i = 2, 3, 4) \tag{12}$$

where $f$ means the fineness ratio of the fuselage and $t/c$ is the ratio of the chord length ($c$) to maximum thickness ($t$) of airfoil. Table 2 shows the aerodynamic characteristic values calculated with respect to form using OpenVSP program.

**Table 2.** HALE solar aircraft wetted and reference area parameters.

| Variable | Fuselage | Main Wing | Horizontal Wing | Vertical Wing |
|---|---|---|---|---|
| Notation ($i$) | 1 | 2 | 3 | 4 |
| Wetted area ($S_{wet}$) | 4.82 | 61.1 | 6.86 | 1.67 |
| Reference area ($S_{ref}$) | 5.44 | 30.3 | 3.36 | 1.92 |
| Form factor ($FF$) | 1.02 | 1.27 | 1.21 | 1.21 |
| $t/c$ | – | 0.13 | 0.10 | 0.10 |
| $f$ | 19.88 | – | – | – |

$C_f$ is the skin friction factor and varies according to the fluid flow (laminar flow, turbulent flow), as in Equation (13). In this study, the laminar flow was assumed in the low Reynolds area [16].

$$C_{f,i} = \begin{cases} \frac{1.328}{\sqrt{Re_i}} & (Laminar) \\ \frac{0.074}{\sqrt{Re^{0.2}}} & (Turbulent) \end{cases} \tag{13}$$

XFLR5 was utilized in this study; aerodynamic analysis was implemented for each main wing and tail wing. Use of low-fidelity methods (XFLR5) can cause aerodynamic modeling errors. However, for fast and low-cost analysis, we used XFLR5 and OpenVSP. The produced aerodynamic data were applied to the self-developed solar aircraft design program. Since the Oswald span coefficient varies depending on the aspect ratio, errors occur in the calculation of the induced drag ($C_{D, induced}$) and the form drag ($C_{D, form}$), by the estimation method. Therefore, XFLR5 is used to increase the accuracy. To understand the aerodynamic characteristics of the previously selected airfoil DAE-11, two-dimensional lift, drag coefficient curve, and drag polar were analyzed as shown in Figure 2.

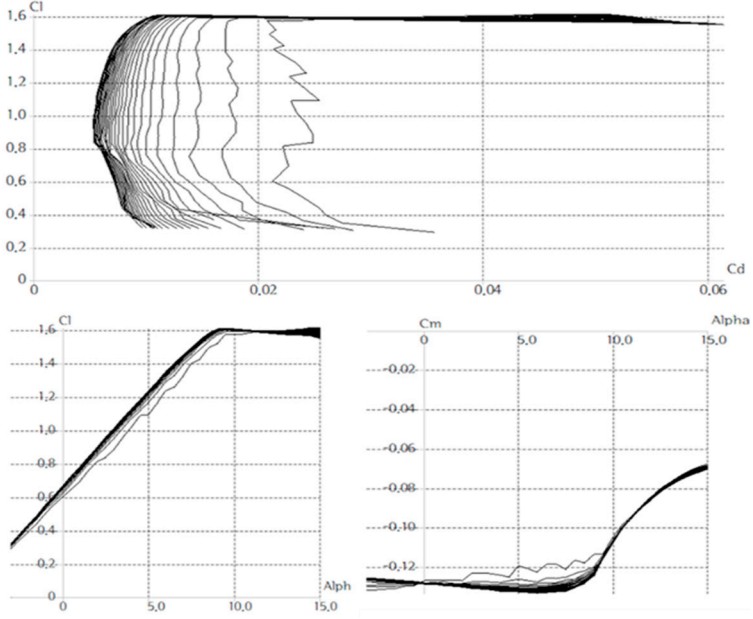

**Figure 2.** DAE-11 airfoil aerodynamic analysis.

Using the aerodynamic analysis results of the DAE-11 airfoil, the vortex lattice method (VLM) for the finite wing (3D) is used to calculate the induced drag ($C_{D,\,induced}$) and the form drag ($C_{D,\,form}$) as shown in Figure 3.

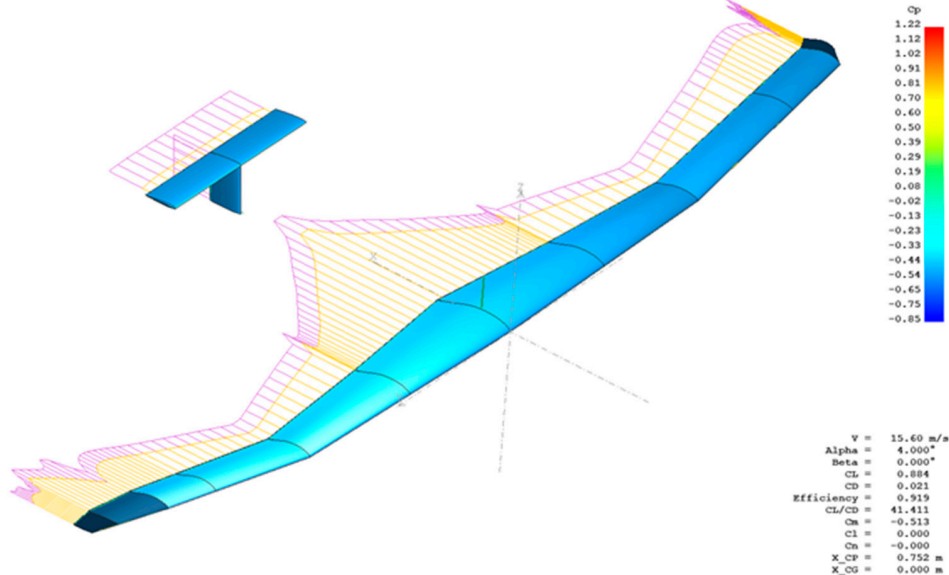

**Figure 3.** SPHALE aircraft aerodynamic analysis using XFLR5.

### 2.3. Reference Flight Condition

Solar aircrafts using solar energy as their source of power are greatly affected by environmental factors during their missions. If the solar energy reaching the earth is regarded as 100%, 49% of it is lost in the troposphere (under 11 km) but 22% is lost in the stratosphere (11–50 km) because of the atmosphere. Therefore, the cruising altitude was set at 18 km, and place of flight was Anheung (latitude: 34.65°, longitude: 126.19°), Republic of Korea.

The solar attenuation factor takes into account the environmental factors such as reflection in the atmosphere or absorption when the light of the sun reaches the earth. This factor varies according to altitude and time of sunrise. As shown in Figure 4, the lower the altitude, the more the attenuation is as the light of the sun passes through the atmosphere, whereas the higher the altitude, the lower the attenuation. The solar attenuation factor were derived from experiments [21] and newly plotted.

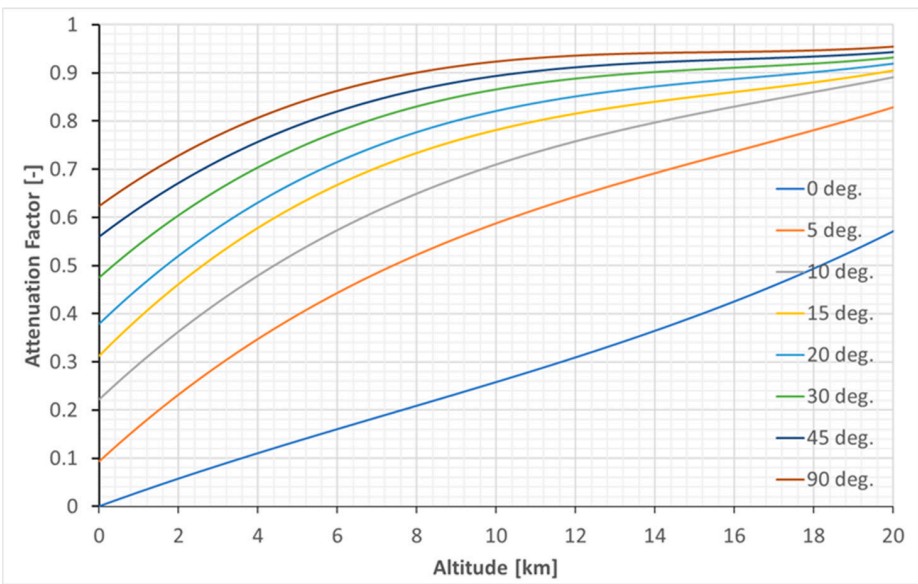

**Figure 4.** Solar attenuation factor.

The date of flight is one of the factors greatly affecting the energy procurable from the sun. In this study, the summer solstice (06/22) was chosen, because this is when the time available to obtain solar energy is the longest in the year. Atmospheric turbidity indicating atmospheric state impacts the solar power per unit area and was set at 100%. The flight conditions and design parameters of the reference model for evaluating flight performance are shown in Table 3.

**Table 3.** Flight condition and design parameters.

| Flight Conditions | Value | Unit |
|---|---|---|
| Flight altitude | 18 | km |
| Flight location | Anheung | – |
| Latitude | 36.45 | deg |
| Longitude | 126.19 | deg |
| Solar attenuation factor | variance | – |
| Flight date | 06/22 | – |
| Flight speed | 18 | m/s |
| Turbidity | 100 | % |
| **Design Parameters** | **Value** | **Unit** |
| Wing span | 24.2 | m |
| Aspect ratio | 18.1 | – |
| Wing area | 30.3 | $m^2$ |
| Payload mass | 5 | kg |
| Payload power | 50 | W |
| Solar cell fill factor | 0.8 | – |
| Solar cell efficiency | 0.2 | – |
| Solar cell specific mass | 0.5 | $kg/m^2$ |
| Battery energy | 10 | kWh |
| Battery specific energy density | 300 | Wh/kg |
| Battery efficiency | 0.9 | – |
| Airframe weight adjustment factor | 1.2 | – |

## 3. Energy Balance Analysis

### 3.1. Required Flight Power

The force on an aircraft during cruise flight is as in Equation (14):

$$L = W$$
$$T = D \tag{14}$$

Since power is defined as the amount of work done per unit time, the required flight power for cruise flight for a day, as in Equation (15), consists of propulsion multiplied by cruise speed and output efficiency.

$$P_{req} = \frac{TV}{\eta_{propulsion}} = \frac{DV}{\eta_{propulsion}} \tag{15}$$

$$\eta_{propulsion} = \eta_{motor}\eta_{propeller} \tag{16}$$

where $\eta_{propulsion}$ represents the efficiency of factor(s) affecting output and is the product of motor efficiency ($\eta_{motor}$) and propeller efficiency ($\eta_{propeller}$).

The total weight utilized in the calculation of required flight power is as in Equation (17):

$$W_T = W_S + W_P + W_{SC} + W_{MP} + W_B \tag{17}$$

where $W_S$ is the structure weight, $W_P$ is the payload weight, $W_{SC}$ is the solar cell weight, $W_{MP}$ is the motor and propeller weight, and $W_B$ is the battery weight.

### 3.2. Weight Estimation

Existing solar aircraft structure weight prediction equations are based on data of actually designed aircraft and empirical equations. D.W. Hall [22] suggested that the sum of all components that make up the airframe structure, such as spar, leading edge, trailing edge, and ribs, is the total weight, but this methodology is very detailed, and there are many variables that must be set initially, which is limited. W. stender [23] proposed a prediction equation using the aspect ratio ($AR$), main wing area ($S$), and the number of boom tails ($n$) as variables, as shown in Equation (18), based on the data of sailplanes with twin boom tails.

$$W_S = 8.763n^{0.311}S^{0.778}AR^{0.467} \tag{18}$$

However, this is optimized for a large scale sailplane, so Andre' Noth proposed a new equation as shown in Equation (19) using the least square fitting method based on the sailplane data [7].

$$W_S = 0.44S^{1.55}AR^{1.3} \tag{19}$$

Usually the solar aircraft weight prediction equation produces excessive prediction than the actual weight since it is applied to the configuration of a large scale sailplane and twin boom tail. Therefore, in this study, a newly derived weight prediction equation was used based on the data of a successful human-powered aircraft. Human-powered aircraft, like solar aircraft, operate in the similar low Reynolds number and have a similar weight range using light-weight materials such as carbon composites. In other words, based on the data of human-powered aircraft, using the aspect ratio and main wing area as variables, a prediction equation can be expressed through regression analysis [24].

$$W_S = -0.0008AR^2 - 0.005S^2 + 0.53AR + 12.88S + 0.027AR{\cdot}S - 10.46 \tag{20}$$

The payload $W_P$ is fixed according to the design requirements, and the solar cell weight $W_{SC}$ can be calculated using Equation (21). $m_{solar}$ is the weight per unit area (kg/m$^2$), $S$ is the wing area (m$^2$), and $S_{ff}$ is the solar cell attachment ratio (%) to the wing area [17].

$$W_{SC} = m_{solar}S_{ff}S \tag{21}$$

Battery weight $W_B$ can be obtained using Equation (22) [17].

$$W_B = \frac{E_{available,B}}{\dot{E}_B} \tag{22}$$

where $E_{available,\ B}$ is the available battery energy (Wh), $\dot{E}_B$ is the battery energy density (Wh/kg).

When the configuration of the referenced Zephyr S is applied, estimated weights of structure, payload, solar cell, motor and propeller, and battery are shown in Table 4. Payload was set to 5 kg, which is the payload value of Zephyr S, and motor and propeller is assumed to be 3 kg. Figure 5 shows the weight breakdown of a reference HALE solar aircraft.

**Table 4.** Initial weight distribution of HALE solar aircraft.

| Variables | Value | Unit |
|---|---|---|
| Structure | 48.9 | kg |
| Payload | 5.0 | kg |
| Solar cell | 12.1 | kg |
| Motor & propeller | 3.0 | kg |
| Battery | 33.3 | kg |
| Total | 102.3 | kg |

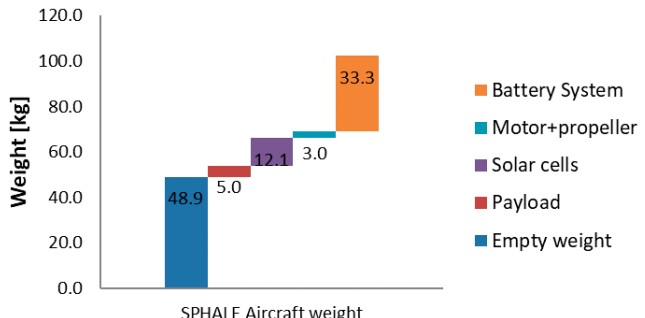

**Figure 5.** Weight breakdown of reference HALE solar aircraft.

### 3.3. Energy Balance Analysis

The energy efficiency of a HALE solar aircraft for a day is shown in Figure 6.

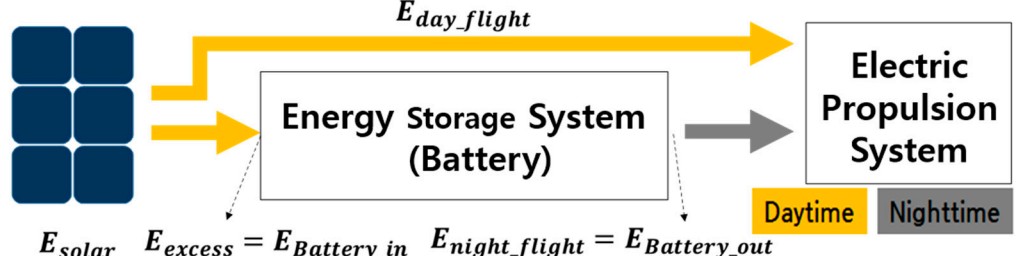

**Figure 6.** Energy flow of HALE solar aircraft.

Solar energy procured during the daytime ($E_{solar}$) is used for flight energy required for daytime flight ($E_{day-flight}$) and remaining excess solar energy ($E_{excess}$) is stored in the battery.

$$E_{excess} = E_{solar} - E_{day-flight} \tag{23}$$

$$E_{Battery-in} = E_{excess} \tag{24}$$

The stored excess solar energy is utilized for flight energy required for nighttime flight ($E_{night-flight}$). $E_{Battery-out}$ is the value after applying battery efficiency. Energy profile for 24 h is shown in Figure 7.

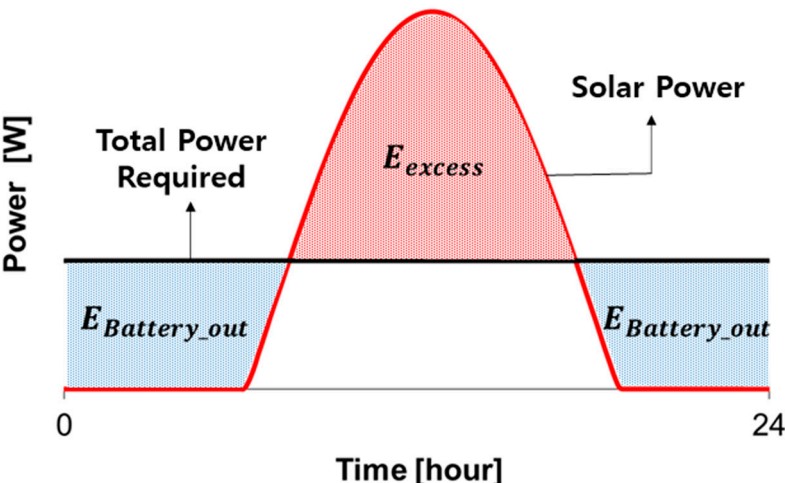

**Figure 7.** Energy balance of HALE solar aircraft.

## 4. General Methodology

To design a HALE solar aircraft in the conceptual design stage, the technology identification, evaluation, and selection (TIES) method is employed. The process is applied in the initial aircraft design

stage for decision-making. The TIES technique is comprehensive and not largely affected by erroneous values. It helps reduce design cost and time by precisely and quickly identifying technologically feasible alternatives, while providing methodological approaches based on diverse probabilistic models such as the response surface method and Monte Carlo simulation [25].

### 4.1. Problem Definition

The first stage of TIES is to define the problem in accordance with the design requirements of the HALE solar aircraft. Since the degree of initial design requirement in the conceptual design phase is found based on the analysis of operation and mission requirements, most of this part is determined by considering the mission analysis. Since the main design requirement for a HALE solar aircraft is long endurance, design parameters such as max take-off weight and wing loading are the primary components. Such design parameters are not fixed in the conceptual design stage but changeable within a specific range until the design requirements are determined [25]. The parameters meeting the requirements are called the design objective parameters. In this study, design objective parameters and specific ranges are set as shown in Table 5.

**Table 5.** Design objective parameters and target values.

| Design Objective Parameter | Target Value | Unit |
|---|---|---|
| Wing loading | ≤30 | N/m$^2$ |
| Power to weight | ≤0.04 | hp/kg |
| Maximum take-off weight | ≤200 | kg |
| Lift to drag ratio | ≥35 | – |

Wing loading is the most important design objective parameter, and the flight speed should be able to exceed the wind speed at the cruising altitude. In consideration of wind speed, the minimum wing loading at about 20 km altitude is approximately 15 N/m$^2$. The range of wing loading for solar aircraft mission implementation at high altitudes is 15~30 N/m$^2$ [26].

The ratio of power to weight is one of the most important factors determining the aircraft performance, together with the wing loading. This design variable has an important impact, particularly on engine and power selection. In this study, as shown in Table 6, the ratio of power to weight ratio for cruise of another solar aircraft was employed. The average values of developed HALE solar aircraft were set as the target values [27]. In this paper, the required power for flight was calculated using the SI unit system, Watt. However, data of the power to weight in other papers were hp/kg so that we converted the unit using the following equation.

$$1 \frac{W}{kg} = \frac{1}{745.7} \frac{hp}{kg} \tag{25}$$

**Table 6.** Power to weight ratios for certain solar aircrafts.

| Solar Aircraft | Value | Unit |
|---|---|---|
| Pathfinder | 0.034 | |
| Pathfinder plus | 0.042 | hp/kg |
| Centurion | 0.036 | |
| Zephyr | 0.045 | |

The maximum take-off weight is one of the factors influencing not only the aircraft performance but also the cost analysis performed using weight data [12]. In this study, 200 kg, the average maximum take-off weight of conventional solar aircraft as shown in Table 7, was used as an initial assumption value.

**Table 7.** Maximum take-off weight of solar aircrafts.

| Solar Aircraft | Value (kg) |
|---|---|
| Solar Riser | 124.7 |
| Solair 1 | 200 |
| Gossamer Penguin | 67.7 |
| Solar Challenger | 153 |
| Pathfinder | 252 |
| Solair 2 | 230 |
| Icare 2 | 360 |
| Solarflugzeug | 280 |
| O sole mio | 220 |
| Pathfinder plus | 315 |
| Average | 220.5 |

Lift-to-drag ratio is a significant indicator representing the performance of a solar aircraft performing a mission at high altitude. The appearance of a HALE aircraft is like a glider or flying wing; this shape is chosen to secure sufficient wing area to generate lift. In the case of a glider form aircraft, the higher the lift-to-drag ratio, the smaller the glide angle and the farther the distance to reach the same altitude. General aviation (GA) aircraft have lift-to-drag ratios of 10~20 in their cruise state. The long distance air vehicle of Virgin Atlantic, the Global Flyer, has a lift-to-drag ratio of 37. Therefore, in this study, the target lift-to-drag ratio was set at 35.

*4.2. Baseline and Alternative Concepts Identification*

There are various subsystems meeting the design requirements in the aircraft conceptual design stage, and comparison analysis is implemented for a large number of combinations. Since the main design requirement for a solar aircraft is long endurance, it is necessary to calculate the required power for a mission and the available power. The main independent design variables necessary for calculating this required power and available power for a solar aircraft are shown in Table 8. As many independent design variables change flexibly in the conceptual design stage as well, calculation should be repeated many times in a specific range in order to find the optimal design conditions [12]. Table 8 shows the major independent design variables and specific ranges of the minimum and maximum values of each variable.

**Table 8.** Define the design space.

| Design Variables | Min | Max | Unit |
|---|---|---|---|
| Wing area | 20 | 50 | $m^2$ |
| Aspect ratio | 10 | 30 | – |
| Flight speed | 18 | 26 | m/s |
| Battery energy | 5 | 10 | kWh |
| Battery specific energy density | 100 | 300 | Wh/kg |
| Battery efficiency | 0.9 | 0.95 | – |
| Solar cell efficiency | 0.15 | 0.24 | – |
| Solar cell specific mass | 0.2 | 0.7 | $kg/m^2$ |
| Solar cell fill factor | 0.78 | 0.8 | – |
| Airframe weight adjustment factor | 0.5 | 1.2 | – |
| Payload weight | 5 | 10 | kg |
| Payload power | 50 | 100 | W |

Since a solar aircraft obtains energy from a solar cell attached to a wing, wing area and aspect ratio are important design variables, and flight speed is also an important design variable to obtain sufficient lift because it operates at high altitude with low atmospheric density. Solar aircraft uses an

electric propulsion system, which uses electric motors and batteries to generate power required for flight. The ultimate goal of solar aircraft, long endurance, depends on the sustainability of night flights. Since the solar energy obtained during the daylight hours is used for daytime flight and the remaining energy is stored in the battery, the main variables of night flight performance are battery energy, battery specific energy density, battery efficiency. Since the main power source is solar energy, the solar cell efficiency, solar cell specific mass, and solar cell fill factor associated with the solar cell are design variables that greatly affect the solar energy calculation. The airframe weight adjustment factor is a coefficient for forcibly adjusting the weight of the structure, and is one of the variables to be considered because the method for calculating the weight of the structure is different depending on the weight estimation formula. In order to overcome over estimating problem, a prediction equation derived from a human-powered aircraft should be modified. This empirical adjustment factor is multiplied to match the structural weight of the human-powered aircraft to the solar aircraft structural weight. Payload weight includes gimbal system equipment such as avionics and cameras, and payload power is an important design variable because it directly affects the flight demand power. Usually, the payload is a given value from customer requirements. However, in this study, we assumed the payload weight can be changed to minimize the total weight.

### 4.3. Modeling and Simulation

In this study, modeling and simulation were performed using the MS Excel program, JMP [15], and a self-developed solar aircraft design framework as shown in Figure 8.

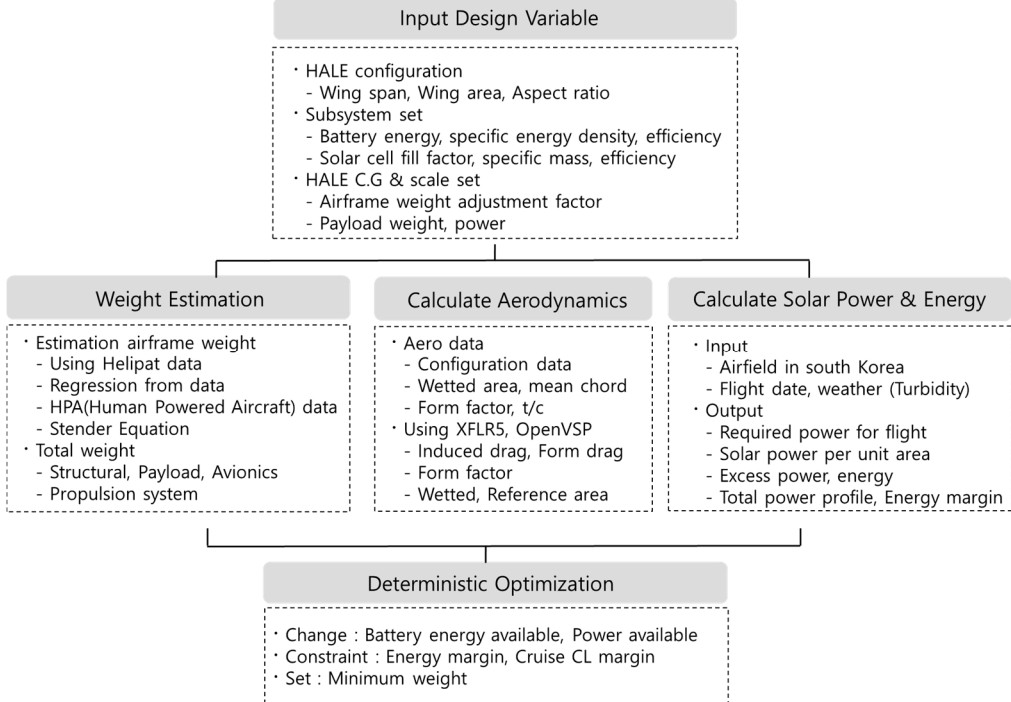

**Figure 8.** Solar aircraft design framework flow chart.

The self-developed solar aircraft design framework conducts initial sizing of solar aircrafts based on Microsoft's Excel, Visual Basic for Application (VBA), and MATLAB.

### 4.4. Design Space Exploration

In the design space exploration phase, the design of experiment (DOE) and screening test technique are employed to check the independent design variables potentially affecting the design objective parameters and to perform the modeling of the relationships among the factors. Then, by utilizing a

regression analysis gained by combining the selected independent design variables, a Monte Carlo simulation is conducted. That is, based on the numerous design variable combinations, a possible design space was found to determine the feasibility of the design objective parameters.

A DOE is a series of tests in which random changes are made to the input variables of a system to observe and identify changes in the output response [28]. In this study, a framework consisting of Excel and VBA is employed. In order to identify variables affecting the performance and the relationships among the factors, design of experiment was established as in Figure 9. The input variables are the main design variables influencing the performance, as defined in Table 8, and the output variables are the aircraft performance-indicating variables, defined in Table 5.

| | Wing area | Aspect ratio | Flight speed | Battery energy | Solar Cell Efficiency | Solar Cell Specific Mass | Battery Specific Energy Density | Solar cell fill factor | Airframe Weight Adjustment Factor | Payload Weight | Payload Power | Battery Efficiency | Wing loading | Power to weight | MTOW | L/D |
|---|---|---|---|---|---|---|---|---|---|---|---|---|---|---|---|---|
| | [m2] | [-] | [m/s] | [kWh] | [-] | [kg/m^2] | [Wh/kg] | [-] | [-] | [kg] | [W] | [-] | [N/m2] | [hp/kg] | [kg] | [-] |
| case1 | 20 | 10 | 18 | 5 | 0.15 | 0.2 | 100 | 0.8 | 1.2 | 5 | 50 | 0.9 | 50.37226 | 0.0193524 | 102.7061 | 30.081834 |
| case2 | 20 | 10 | 18 | 5 | 0.24 | 0.7 | 100 | 0.78 | 0.5 | 10 | 100 | 0.9 | 50.69211 | 0.0194 | 103.3663 | 30.11228 |
| case3 | 20 | 10 | 18 | 5 | 0.15 | 0.7 | 100 | 0.78 | 1.2 | 10 | 50 | 0.95 | 60.22001 | 0.020884 | 122.7929 | 30.55004 |
| case4 | 20 | 10 | 18 | 5 | 0.24 | 0.2 | 100 | 0.8 | 0.5 | 5 | 100 | 0.95 | 35.06528 | 0.0166198 | 71.49665 | 28.294432 |
| case5 | 20 | 10 | 18 | 5 | 0.15 | 0.7 | 300 | 0.8 | 0.5 | 10 | 50 | 0.9 | 44.51023 | 0.0183948 | 90.762 | 29.347051 |
| ⋮ | ⋮ | ⋮ | ⋮ | ⋮ | ⋮ | ⋮ | ⋮ | ⋮ | ⋮ | ⋮ | ⋮ | ⋮ | ⋮ | ⋮ | ⋮ | ⋮ |
| case26 | 20 | 10 | 26 | 10 | 0.24 | 0.7 | 100 | 0.8 | 1.2 | 10 | 100 | 0.9 | 67.13693 | 0.0218713 | 136.8966 | 31.092232 |
| case27 | 20 | 10 | 26 | 10 | 0.15 | 0.7 | 100 | 0.8 | 0.5 | 10 | 50 | 0.95 | 44.58501 | 0.0184077 | 90.91446 | 29.333246 |
| case28 | 20 | 10 | 26 | 10 | 0.24 | 0.2 | 100 | 0.78 | 1.2 | 5 | 100 | 0.95 | 54.07216 | 0.0199349 | 110.2501 | 30.10816 |
| case29 | 20 | 10 | 26 | 10 | 0.15 | 0.7 | 300 | 0.78 | 1.2 | 10 | 50 | 0.9 | 60.21519 | 0.0208833 | 122.7831 | 30.545729 |
| case30 | 20 | 10 | 26 | 10 | 0.24 | 0.2 | 300 | 0.8 | 0.5 | 5 | 100 | 0.9 | 19.56136 | 0.0129752 | 39.88596 | 26.332974 |
| case31 | 20 | 10 | 26 | 10 | 0.15 | 0.2 | 300 | 0.8 | 1.2 | 5 | 50 | 0.95 | 50.25268 | 0.0193339 | 102.4622 | 30.052347 |
| case32 | 20 | 10 | 26 | 10 | 0.24 | 0.7 | 300 | 0.78 | 0.5 | 10 | 100 | 0.95 | 40.75061 | 0.0177249 | 83.09639 | 28.926951 |
| ⋮ | ⋮ | ⋮ | ⋮ | ⋮ | ⋮ | ⋮ | ⋮ | ⋮ | ⋮ | ⋮ | ⋮ | ⋮ | ⋮ | ⋮ | ⋮ | ⋮ |
| case120 | 50 | 30 | 26 | 5 | 0.24 | 0.2 | 300 | 0.78 | 1.2 | 10 | 50 | 0.95 | 26.69401 | 0.0087205 | 136.0673 | 44.167603 |
| case121 | 50 | 30 | 26 | 10 | 0.15 | 0.2 | 100 | 0.8 | 1.2 | 5 | 50 | 0.9 | 52.51792 | 0.0114247 | 267.6944 | 48.308147 |
| case122 | 50 | 30 | 26 | 10 | 0.24 | 0.7 | 100 | 0.78 | 0.5 | 10 | 100 | 0.9 | 45.8662 | 0.0108119 | 233.8097 | 47.470896 |
| case123 | 50 | 30 | 26 | 10 | 0.15 | 0.7 | 100 | 0.78 | 1.2 | 10 | 50 | 0.95 | 62.9145 | 0.0122691 | 320.7067 | 49.659939 |
| case124 | 50 | 30 | 26 | 10 | 0.24 | 0.2 | 100 | 0.8 | 0.5 | 5 | 100 | 0.95 | 34.58171 | 0.0096736 | 176.2719 | 45.762569 |
| case125 | 50 | 30 | 26 | 10 | 0.15 | 0.7 | 300 | 0.8 | 0.5 | 10 | 50 | 0.9 | 18.51322 | 0.0074893 | 94.39012 | 42.620679 |
| case126 | 50 | 30 | 26 | 10 | 0.24 | 0.2 | 300 | 0.78 | 1.2 | 5 | 100 | 0.9 | 25.76702 | 0.0085949 | 131.3427 | 43.981783 |
| case127 | 50 | 30 | 26 | 10 | 0.15 | 0.2 | 300 | 0.78 | 0.5 | 5 | 50 | 0.95 | 11.66354 | 0.0062182 | 59.45665 | 40.090303 |
| case128 | 50 | 30 | 26 | 10 | 0.24 | 0.7 | 300 | 0.8 | 1.2 | 10 | 100 | 0.95 | 33.73479 | 0.009579 | 171.9755 | 45.585906 |

Input variables · Output variables

**Figure 9.** Design of experiment in HALE framework.

### 4.4.1. Screening Test

A screening test is an experiment-designing method utilized to find variables with the greatest effect on the design objective parameters. In this study, with respect to the levels of factors, 2 factors were utilized, so that there were only max and min values. Input variable combinations, as shown in Figure 10, are generated using the JMP program. The table on the left in the figure was used to form an experimental design using fractional factorial design. As explained above, −1 and 1 indicate 2 factor level. The table on the right shows values converted to the actual scales. In the fractional factorial design, −1 represents the actual min; and 1, the actual max values.

These design variable combinations created using the fractional factorial design and their performance indicators were used to perform regression analysis. The resulting graph is shown in Figure 11. In the graph, the *x*-axis and *y*-axis represent the predicted value and the actual value respectively. The actual value is the output variables calculated using the HALE solar framework. Regression equations are derived by modeling the relationship between the input variables and the calculated output variables. The predicted value is the value calculated using the regression equations. Each small dot in the graph means 128 cases; the red-colored dotted line shows how well the model was estimated. The red line represents a perfect model. The closer the dotted line is to this red line, the better the fitting is. However, since the screening test, using a linear DOE, is performed to find variables with the largest effect on the performance, the distances between two dotted lines seem large. In addition, the determination coefficient, ($R^2$), is one of the indicators that can be used to evaluate the optimized response surface equations (RSE) and represents the data variance ratio as an indicator showing how well the regression function model fits the estimations. Determination coefficient values are always

between 0 and 1; the closer to 1, the better the explanation of the relationship between the design objective parameters and the independent variables will be. As shown in Table 9, the determination coefficient is also unsatisfactory. This is compared with the results of regression analysis performed using the response surface method after selecting independent variables with the largest effect on the design objective variables through pareto plot and *p*-value.

| | Pattern | Wing area | Aspect ratio | Flight speed | | Wing area [m2] | Aspect ratio [-] | Flight speed [m/s] |
|---|---|---|---|---|---|---|---|---|
| 1 | – – – – – ... | -1 | -1 | -1 | | 20 | 10 | 18 |
| 2 | – – – – – ... | -1 | -1 | -1 | | 20 | 10 | 18 |
| 3 | – – – – – ... | -1 | -1 | -1 | | 20 | 10 | 18 |
| 4 | – – – – – ... | -1 | -1 | -1 | | 20 | 10 | 18 |
| 5 | – – – – + ... | -1 | -1 | -1 | | 20 | 10 | 18 |
| 6 | – – – – + ... | -1 | -1 | -1 | | 20 | 10 | 18 |
| 7 | – – – – + ... | -1 | -1 | -1 | | 20 | 10 | 18 |
| 8 | – – – – + ... | -1 | -1 | -1 | | 20 | 10 | 18 |
| 9 | – – – + – ... | -1 | -1 | -1 | | 20 | 10 | 18 |
| 10 | – – – + – ... | -1 | -1 | -1 | | 20 | 10 | 18 |
| 11 | – – – + – ... | -1 | -1 | -1 | | 20 | 10 | 18 |
| 12 | – – – + – ... | -1 | -1 | -1 | | 20 | 10 | 18 |
| 13 | – – – + + ... | -1 | -1 | -1 | | 20 | 10 | 18 |
| 14 | – – – + + ... | -1 | -1 | -1 | | 20 | 10 | 18 |
| 15 | – – – + + ... | -1 | -1 | -1 | | 20 | 10 | 18 |
| 16 | – – – + + ... | -1 | -1 | -1 | | 20 | 10 | 18 |
| 17 | – – + – – ... | -1 | -1 | 1 | | 20 | 10 | 26 |

**Figure 10.** Conversion of actual design variable values by fractional factorial design.

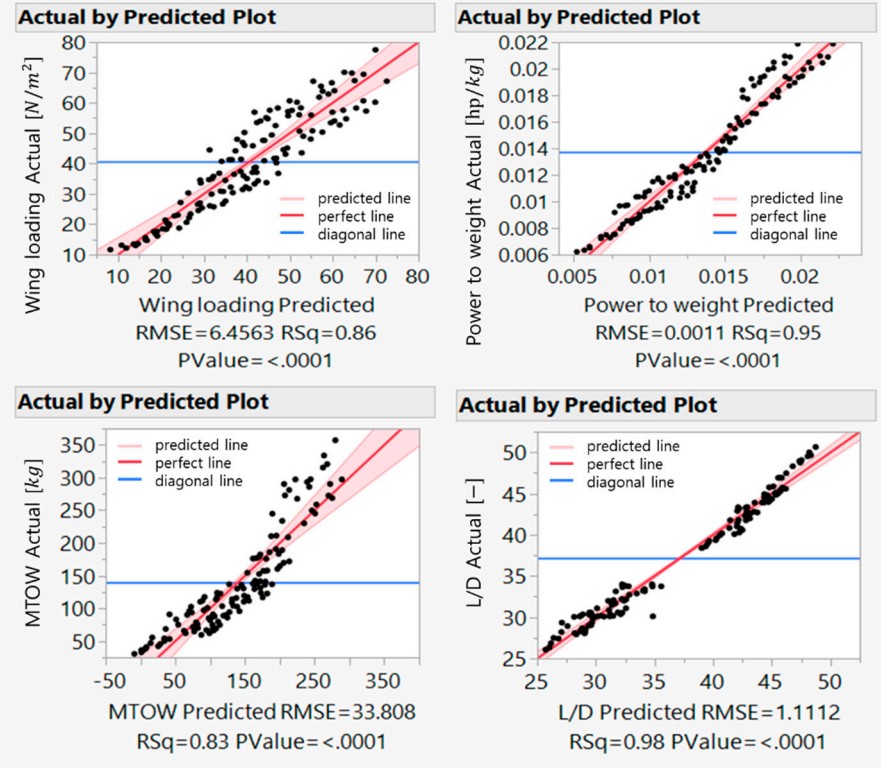

**Figure 11.** Actual predicted plots of design target variables.

**Table 9.** $R^2$ values of screening test.

| Classification | $R^2$ |
|---|---|
| Wing loading | 0.86 |
| Power to weight | 0.95 |
| Maximum take-off weight | 0.83 |
| Lift to drag ratio | 0.98 |

After this, to determine the independent design variables with the largest influence on the design objective parameters, a pareto graph was analyzed. Pareto graphs express responses to individual independent design variables. The lines in each graph show the accumulated effect of the independent design variables [25]. As can be seen in Figure 12, the airframe weight adjustment factor and the battery-specific energy density are the two primary contributors to the wing loading, while the solar cell fill factor and the battery energy hardly contribute at all. The pareto plot is a means to visually determine the most significant contributors to a response, and a parameter estimate can be used to numerically determine the important variables.

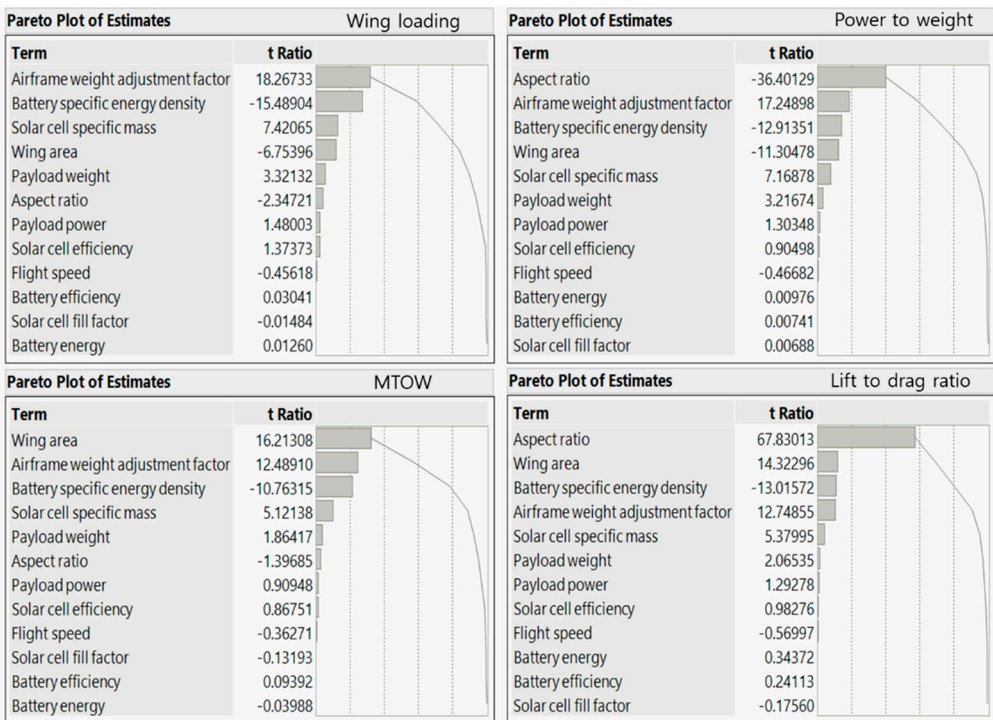

**Figure 12.** Pareto plot of design target variables.

In Figure 13, looking at "Prob > |t|" in the second column, if this value is less than 0.05, the parameter estimate significantly influences the response. As shown in Table 10, the most influential independent design variables can be selected by comparing the pareto plot and the independent design variables with a parameter estimate of less than 0.05.

### 4.4.2. Response Surface Equations

The response surface method uses mathematical and statistical techniques to build an empirical model, and optimizes the response of various independent variables through design experiments. As in the equation below, optimization is performed using the second model:

$$R = b_0 + \sum_{i=1}^{k} b_i k_i + \sum_{i=1}^{k} b_{ii} k_i^2 + \sum_{j=1}^{k-1} \sum_{j=i+1}^{k} b_{ij} k_i k_j \tag{26}$$

where $b_i$ is a regression coefficient for linear terms, $b_{ii}$ is a coefficient for pure quadratic terms, $b_{ij}$ is a coefficient for cross-product terms, $k_i$ and $k_j$ are the design variables, and $k_i k_j$ denotes the interactions between the two design variables [25].

| Parameter Estimates | Wing loading | | Parameter Estimates | Power to weight | |
|---|---|---|---|---|---|
| **Term** | **Estimate** | **Prob>\|t\|** | **Term** | **Estimate** | **Prob>\|t\|** |
| Intercept | 26.904592 | 0.5923 | Intercept | 0.0188071 | 0.0239* |
| Wing area | -0.256947 | <.0001* | Wing area | -7.053e-5 | <.0001* |
| Aspect ratio | -0.133945 | 0.0206* | Aspect ratio | -0.000341 | <.0001* |
| Flight speed | -0.065081 | 0.6491 | Flight speed | -0.000011 | 0.6415 |
| Battery energy | 0.0028761 | 0.9900 | Battery energy | 3.6522e-7 | 0.9922 |
| Battery specific energy density | -0.08839 | <.0001* | Battery specific energy density | -0.000012 | <.0001* |
| Battery efficiency | 0.694127 | 0.9758 | Battery efficiency | 2.774e-5 | 0.9941 |
| Solar cell efficiency | 17.420644 | 0.1722 | Solar cell efficiency | 0.001882 | 0.3674 |
| Solar cell specific mass | 16.938644 | <.0001* | Solar cell specific mass | 0.0026835 | <.0001* |
| Solar cell fill factor | -0.846722 | 0.9882 | Solar cell fill factor | 6.434e-5 | 0.9945 |
| Airframe weight adjustment factor | 29.784027 | <.0001* | Airframe weight adjustment factor | 0.0046121 | <.0001* |
| Payload weight | 0.7581364 | 0.0012* | Payload weight | 0.0001204 | 0.0017* |
| Payload power | 0.0337837 | 0.1416 | Payload power | 4.8794e-6 | 0.1950 |

| Parameter Estimates | MTOW | | Parameter Estimates | Lift to drag ratio | |
|---|---|---|---|---|---|
| **Term** | **Estimate** | **Prob>\|t\|** | **Term** | **Estimate** | **Prob>\|t\|** |
| Intercept | -28.33197 | 0.9142 | Intercept | 18.445487 | 0.0346* |
| Wing area | 3.2298581 | <.0001* | Wing area | 0.0937824 | <.0001* |
| Aspect ratio | -0.417405 | 0.1651 | Aspect ratio | 0.6661966 | <.0001* |
| Flight speed | -0.270964 | 0.7175 | Flight speed | -0.013995 | 0.5698 |
| Battery energy | -0.047663 | 0.9683 | Battery energy | 0.0135033 | 0.7317 |
| Battery specific energy density | -0.321624 | <.0001* | Battery specific energy density | -0.012783 | <.0001* |
| Battery efficiency | 11.225788 | 0.9253 | Battery efficiency | 0.9473251 | 0.8099 |
| Solar cell efficiency | 57.606565 | 0.3875 | Solar cell efficiency | 2.1449297 | 0.3278 |
| Solar cell specific mass | 61.214806 | <.0001* | Solar cell specific mass | 2.1135763 | <.0001* |
| Solar cell fill factor | -39.42273 | 0.8953 | Solar cell fill factor | -1.724666 | 0.8609 |
| Airframe weight adjustment factor | 106.62827 | <.0001* | Airframe weight adjustment factor | 3.577442 | <.0001* |
| Payload weight | 2.2282017 | 0.0648 | Payload weight | 0.0811398 | 0.0411* |
| Payload power | 0.1087085 | 0.3650 | Payload power | 0.0050789 | 0.1987 |

**Figure 13.** *p*-values of design target variables. * denotes values less than 0.05.

**Table 10.** Selected independent design variables.

| Classification | Unit |
|---|---|
| Aspect ratio (X1) | – |
| Wing area (X2) | $m^2$ |
| Airframe weight adjustment factor (X3) | – |
| Battery specific energy density (X4) | Wh/kg |
| Solar cell specific mass (X5) | $kg/m^2$ |
| Solar cell efficiency (X6) | – |
| Payload weight (X7) | kg |

The central composition design is used to construct an optimization model for the response surface method. The central composition design is a method of minimizing the number of experiments to optimize the second-order model, and consists of $2^n$ factorial points, 2n axis points, and one center point. In this study, the central composition is designed using the JMP, with 144 cases and 2 center points [12]. Then, as in the screening test, design of experiment is performed to derive the performance variables for each case. The graph of the regression analysis results compares the values calculated via the regression equation with the actual values, as shown in Figure 14. The determination coefficient ($R^2$), as shown in Table 11, had a value close to 1, indicating that the relationship between the design objective parameters and the independent variables was well explained. The regression analysis results under the response surface method using the second model, in comparison with the screening test results using the first model, are shown.

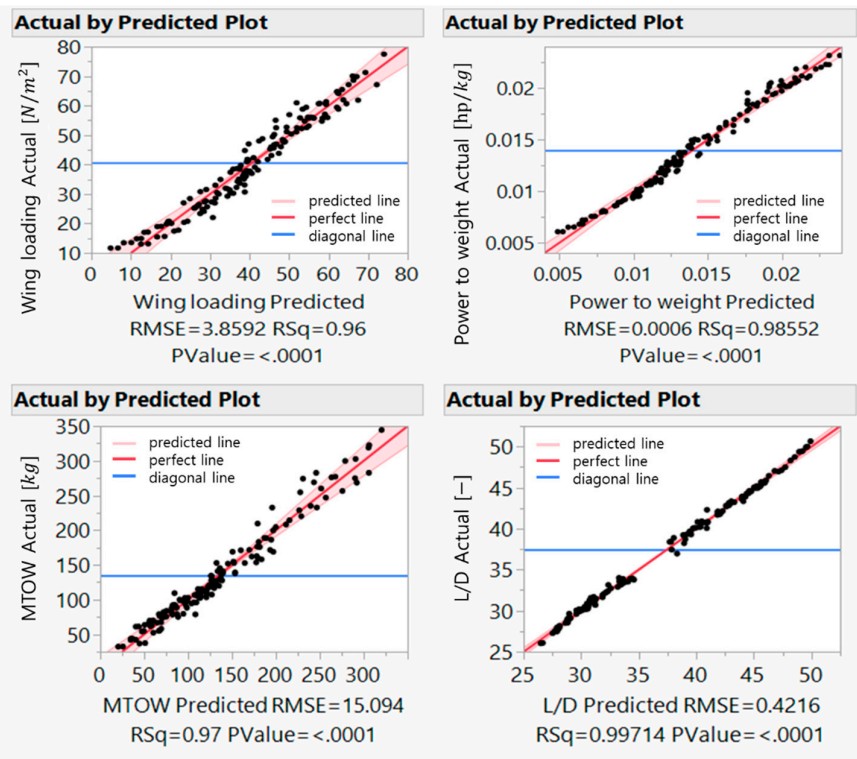

**Figure 14.** Regression results of design target variables.

**Table 11.** $R^2$ values of response surface equations.

| Classification | $R^2$ |
|---|---|
| Wing loading | 0.96 |
| Power to weight | 0.98 |
| Maximum take-off weight | 0.97 |
| Lift to drag ratio | 0.99 |

Figure 15 shows a surrogate model indicating the relationship between the selected independent design variables and the design objective parameters. The horizontal axis represents the seven selected independent design variables; the vertical axis shows the design objective parameters indicating aircraft performance. Each inclination is sensitive to the independent variables. The larger the inclination, the higher the sensitivity. For instance, when the aspect ratio (X1) rises, the power to weight falls sharply, but the lift-to-drag ratio moves up very much. Moreover, the relationship between solar cell efficiency (X6) and lift to drag ratio has almost 0 inclination, signaling that the solar cell efficiency, an independent design variable, has little impact on the lift-to-drag ratio.

### 4.4.3. Monte Carlo Simulation

In the aircraft conceptual design phase, a stochastic model is employed that does not provide precise result prediction. Unlike deterministic models, which predict results precisely with clear variable relationships, the stochastic model makes it impossible to find an analytical solution. To resolve this problem, a series of numerically random numbers are repeatedly generated to perform simulation in a technique called Monte Carlo simulation.

The Monte Carlo simulation analyzes the stochastic model of unit variables and estimates a model of unit variable combinations. That is, by integrating the probability distribution of variables, it finds the probability distribution of a target value. In this study, random numbers were created within a set design range and the Monte Carlo simulation was implemented using the model produced through the response surface equation.

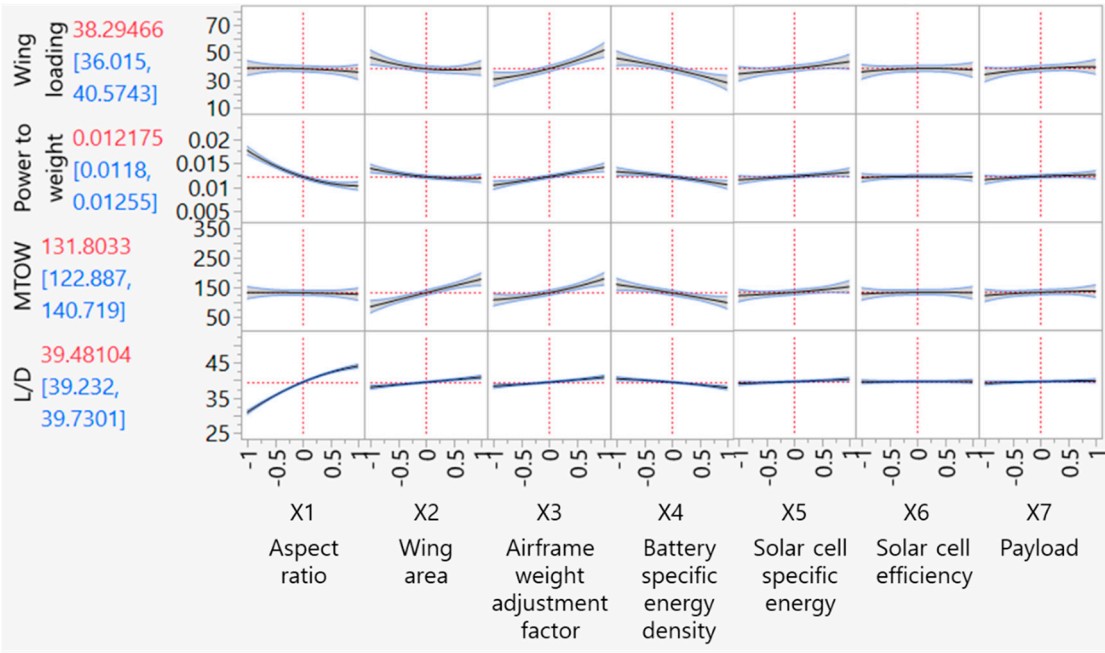

**Figure 15.** Prediction profile for selected design objective variables.

Based on the response surface method performed above, a regression equation can be established for the independent design variables and their combinations. Regression coefficients are organized in Table 12. The equation established from the regression analysis can to a great extent save time required for design of experiment for screening test and response surface method, contributing to an efficient determination of the overall design point. For example, the following equation is a regression equation for the wing loading; the intercept is the y-intercept. $X_1 \sim X_7$ are the independent design variables selected using the screening test above, and $\beta_1 \sim \beta_{35}$ are the regression coefficients for each term. The following is an expression representing various combinations of selected independent design variables; the regression coefficient is the effect of the combination.

$$\text{Wing loading} = \text{Intercept} + X_1\beta_1 + X_2\beta_2 + \cdots + X_7\beta_7 + X_1X_2\beta_8 + X_1X_3\beta_9 + \cdots + X_7X_7\beta_{35} \quad (27)$$

**Table 12.** Regression coefficients for response surface method.

| Term | Coefficient (β) | Wing Loading | Power to Weight | MTOW | L/D |
|------|-----------------|--------------|-----------------|------|-----|
| Intercept | | 38.29466 | 0.012175 | 131.8033 | 39.48104 |
| X1 | 1 | −1.45885 | −0.00369 | −2.72136 | 6.619312 |
| X2 | 2 | −3.8979 | −0.00103 | 46.53446 | 1.432357 |
| X3 | 3 | 10.59271 | 0.001857 | 35.43302 | 1.300914 |
| X4 | 4 | −8.89292 | −0.00136 | −30.7049 | −1.3019 |
| X5 | 5 | 4.420539 | 0.000786 | 14.93174 | 0.583766 |
| X6 | 6 | 0.719599 | $9.49 \times 10^{-5}$ | 2.12396 | 0.057537 |
| X7 | 7 | 2.764475 | 0.000503 | 7.604405 | 0.333755 |
| X1 × X2 | 8 | 0.445333 | 0.000299 | −0.02664 | 0.180061 |
| X1 × X3 | 9 | −1.26241 | −0.00067 | −3.79547 | 0.04719 |
| X2 × X3 | 10 | 0.097532 | $3.18 \times 10^{-5}$ | 15.38417 | 0.206605 |
| X1 × X4 | 11 | −4.86717 | −0.00037 | −15.8155 | −0.7403 |
| X2 × X4 | 12 | −0.48553 | −0.00011 | −14.4865 | −0.29344 |
| X3 × X4 | 13 | 0.682484 | 0.000418 | 1.828877 | 0.294346 |
| X1 × X5 | 14 | −0.59253 | −0.0003 | −1.91813 | 0.032379 |
| X2 × X5 | 15 | 0.144814 | $2.26 \times 10^{-5}$ | 6.716138 | 0.113982 |
| X3 × X5 | 16 | 0.109353 | −0.00011 | 0.962494 | −0.10835 |

**Table 12.** *Cont.*

| Term | Coefficient (β) | Wing Loading | Power to Weight | MTOW | L/D |
|---|---|---|---|---|---|
| X4 × X5 | 17 | 0.160074 | 0.000151 | 0.533371 | 0.116464 |
| X1 × X6 | 18 | −0.01231 | $-3.18 \times 10^{-5}$ | 0.26585 | 0.014945 |
| X2 × X6 | 19 | −0.17732 | $-3.23 \times 10^{-5}$ | 0.505359 | −0.00048 |
| X3 × X6 | 20 | 0.393237 | $5.18 \times 10^{-5}$ | 0.949667 | 0.02179 |
| X4 × X6 | 21 | −0.84886 | −0.00013 | −2.82743 | −0.09791 |
| X5 × X6 | 22 | 0.231659 | $3.54 \times 10^{-5}$ | 0.919121 | 0.015236 |
| X1 × X7 | 23 | −0.19177 | −0.00016 | −0.40344 | 0.034217 |
| X2 × X7 | 24 | −1.05965 | −0.0002 | 0.402157 | −0.08273 |
| X3 × X7 | 25 | −0.17949 | −0.00012 | −0.21023 | −0.07678 |
| X4 × X7 | 26 | −0.01868 | $8.06 \times 10^{-5}$ | −0.18041 | 0.036426 |
| X5 × X7 | 27 | −0.00588 | $-2.63 \times 10^{-5}$ | −0.37093 | −0.02592 |
| X6 × X7 | 28 | −0.0146 | $-1.07 \times 10^{-5}$ | −0.14558 | −0.00737 |
| X1 × X1 | 29 | −0.93147 | 0.001789 | −2.10506 | −2.01678 |
| X2 × X2 | 30 | 4.414936 | 0.000694 | −0.72646 | −0.008 |
| X3 × X3 | 31 | 3.035355 | $8.08 \times 10^{-5}$ | 11.95347 | 0.14692 |
| X4 × X4 | 32 | −1.32252 | −0.0003 | −2.47769 | −0.31106 |
| X5 × X5 | 33 | 0.562144 | $5.11 \times 10^{-5}$ | 3.928954 | 0.043915 |
| X6 × X6 | 34 | −1.70642 | −0.0002 | −3.59658 | 0.006487 |
| X7 × X7 | 35 | −1.58519 | −0.00019 | −3.21738 | −0.10288 |

After that, random numbers between −1 and 1 were generated 10,000 times using the Excel Random function. The design objective parameter values were calculated for the 10,000 numbers using the regression equation.

As shown in Figure 16, the graphs resulting from the Monte Carlo simulation represent the frequency of 10,000 design objective parameters and are a stochastic model built in consideration of various factors influencing each objective variable.

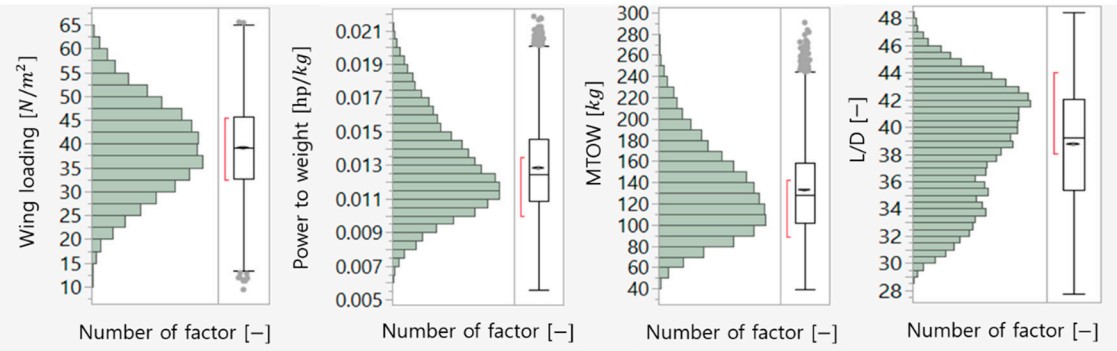

**Figure 16.** Independent design variables distribution.

*4.5. Determination of System Feasibility/Viability*

Based on the Monte Carlo simulation results, the design feasibility of the design objective parameters was assessed. The design feasibility was determined through the cumulative distribution function (CDF). This shows the design results of every geometric combination of the ranges of the independent design variables.

4.5.1. Cumulative Distribution Function

The cumulative distribution function is utilized to express the probability distribution of the stochastic variable, $x$, which has a real number value. The reason for including the word cumulative, is to signify the integral value of the probability density function. In other words, point probability can be expressed as in the equation below. Since the cumulative distribution function is an increasing function, it is easy to see the probability of it being smaller than or the same as a certain value.

As shown in Figure 17, the percentiles are indications showing the distribution state of the cumulative distribution function.

$$P(X = b) = F(b) - \lim_{x \to b^-} F(x) \tag{28}$$

| Quantiles | | Wing loading | Quantiles | | Power to weight | Quantiles | | MTOW | Quantiles | | Lift to drag ratio |
|---|---|---|---|---|---|---|---|---|---|---|---|
| 100.0% | maximum | 65.53440002 | 100.0% | maximum | 0.021836044 | 100.0% | maximum | 290.547822 | 100.0% | maximum | 48.38010885 |
| 99.5% | | 61.134557301 | 99.5% | | 0.0202460757 | 99.5% | | 248.20036656 | 99.5% | | 46.755390396 |
| 97.5% | | 57.381291465 | 97.5% | | 0.0187719499 | 97.5% | | 223.54293814 | 97.5% | | 45.522967528 |
| 90.0% | | 51.375175476 | 90.0% | | 0.0167250894 | 90.0% | | 189.79924976 | 90.0% | | 43.863776433 |
| 75.0% | quartile | 45.754513878 | 75.0% | quartile | 0.0145730405 | 75.0% | quartile | 158.85560693 | 75.0% | quartile | 42.035886148 |
| 50.0% | median | 39.18924047 | 50.0% | median | 0.0124562515 | 50.0% | median | 127.867966 | 50.0% | median | 39.210896765 |
| 25.0% | quartile | 32.669947435 | 25.0% | quartile | 0.010871296 | 25.0% | quartile | 101.89803523 | 25.0% | quartile | 35.393570655 |
| 10.0% | | 27.081450143 | 10.0% | | 0.0096241012 | 10.0% | | 84.068874304 | 10.0% | | 32.690138026 |
| 2.5% | | 21.430960666 | 2.5% | | 0.0082376672 | 2.5% | | 67.469274719 | 2.5% | | 30.828709461 |
| 0.5% | | 17.245262727 | 0.5% | | 0.0072665769 | 0.5% | | 54.092566843 | 0.5% | | 29.761676338 |
| 0.0% | minimum | 9.42148976 | 0.0% | minimum | 0.005557752 | 0.0% | minimum | 38.83081546 | 0.0% | minimum | 27.72416743 |

**Figure 17.** Percentiles for design objective variables.

Wing loading was found to be between 9.40 N/m² and 65.5 N/m², and the targeted wing loading value is 30 N/m² or under. Therefore, the design probability to satisfy wing loading is 20%. Power to weight ratio was found to be from 0.005 hp/kg to 0.022 hp/kg. Since the targeted power to weight ratio is 0.04 hp/kg or lower, the design probability is 100%. Maximum take-off weight was between 38 kg and 291 kg, and the targeted max take-off weight is 200 kg or lower. Thus, the design probability to meet this maximum take-off weight is 92.4%. Lift-to-drag ratio was from 27.72 to 48.38. The target lift-to-drag ratio is 35 or higher, so the design probability to meet this is 75%. The design feasibility of every design objective variable was assessed, and it was found that the design objective variables can be satisfied within the independent design variable ranges set previously.

Figure 18 provides graphs of the percentages according to the designable ranges of the design objective parameters. The horizontal axis provides the design objective parameter values produced in the Monte Carlo simulation; the vertical axis shows the percentages representing feasibility. For instance, in the case of wing loading, the targeted value is 30 N/m²; thus, accordingly, the design probability is found to be 20% on the vertical axis. As for the maximum take-off weight, its target value is 200 kg, and, the design probability is 92.4% along the vertical axis. In this manner, it is found that these values are the same as the design probability values analyzed in the percentile table above.

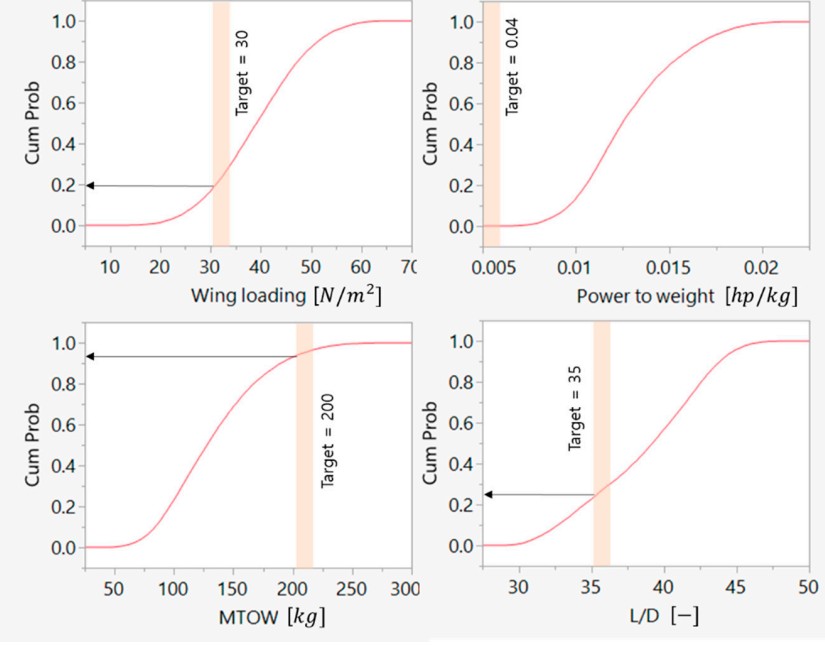

**Figure 18.** Cumulative probabilities for design objective variables.

### 4.5.2. Desirability Function for Optimization

Optimization of HALE solar aircraft was performed using desirability function based on the results of response surface methodology. The desirability function is one of the most popular performance metrics for simultaneous optimization of multiple continuous response variables [29]. The individual responses are first estimated through regressions, and then each estimated response is converted to the desirability value. The individual desirabilities are then combined using the geometric mean to get the overall desirability [29]. The desirability functions are categorized as nominal-the-better (NTB), smaller-the-better (STB), and larger-the-better (LTB), as shown in Equations (29)–(31) as follows [12]:

$$NTB : d_i = \begin{cases} \left(\frac{\hat{y}_i - y_i^L}{T_i - y_i^L}\right)^s & if \ y_i^L \le \hat{y}_i \le T_i \\ \left(\frac{\hat{y}_i - y_i^U}{T_i - y_i^U}\right)^t & if \ T_i < \hat{y}_i \le y_i^U \\ 0 & if \ \hat{y}_i > y_i^U \end{cases} \tag{29}$$

$$STB : d_i = \begin{cases} 1 & if \ \hat{y}_i < T_i \\ \left(\frac{\hat{y}_i - y_i^U}{T_i - y_i^U}\right)^s & if \ T_i \le \hat{y}_i \le y_i^U \\ 0 & if \ \hat{y}_i > y_i^U \end{cases} \tag{30}$$

$$LTB : d_i = \begin{cases} 0 & if \ \hat{y}_i < y_i^L \\ \left(\frac{\hat{y}_i - y_i^L}{T_i - y_i^L}\right)^s & if \ y_i^L < \hat{y}_i < T_i \\ 1 & if \ \hat{y}_i > T_i \end{cases} \tag{31}$$

where $d_i$ is the desirability value, $\hat{y}_i$ is the desired response, $y_i^L$ is the lower value, and $y_i^U$ is the upper value, $T_i$ is the target value. The exponents $s$ and $t$ are the shape constants of the desirability function, in general, they are chosen in the range from 0.01 to 10. The $y_i^L$ and $y_i^U$ are respectively the lower and upper specification limit for each type response variable with a target value $T_i$ [30].

The goal of the desirability function approach is to find the combination of design variables by which the geometric mean is maximized for each desirability function. The geometric mean is defined as shown in Equation (32). The function shape can be determined by changing the variables. Once these variables are defined as greater than 1, they approach the target values [29]:

$$D = (d_1 \times d_2 \times \cdots \times d_n)^{\frac{1}{n}} \tag{32}$$

In this study, the smaller-the-better (STB) optimization was used to optimize the design objective parameters, wing loading, power to weight, maximum take-off weight defined in Table 5. Lift to drag ratio was optimized using the larger-the-better (LTB) optimization through JMP "desirability function." In Figure 19, the *x*-axis represents the independent design variables and the *y*-axis represents the response variables. The desirability characteristic is presented by the slope on the right. The smaller-the-better (STB) has a negative slope whereas the larger-the-better (LTB) has a positive slope.

As shown in Table 13, the optimized independent design variable values are converted to actual values, and these values can be applied to the solar aircraft design framework to derive design objective parameter values as shown in Table 14.

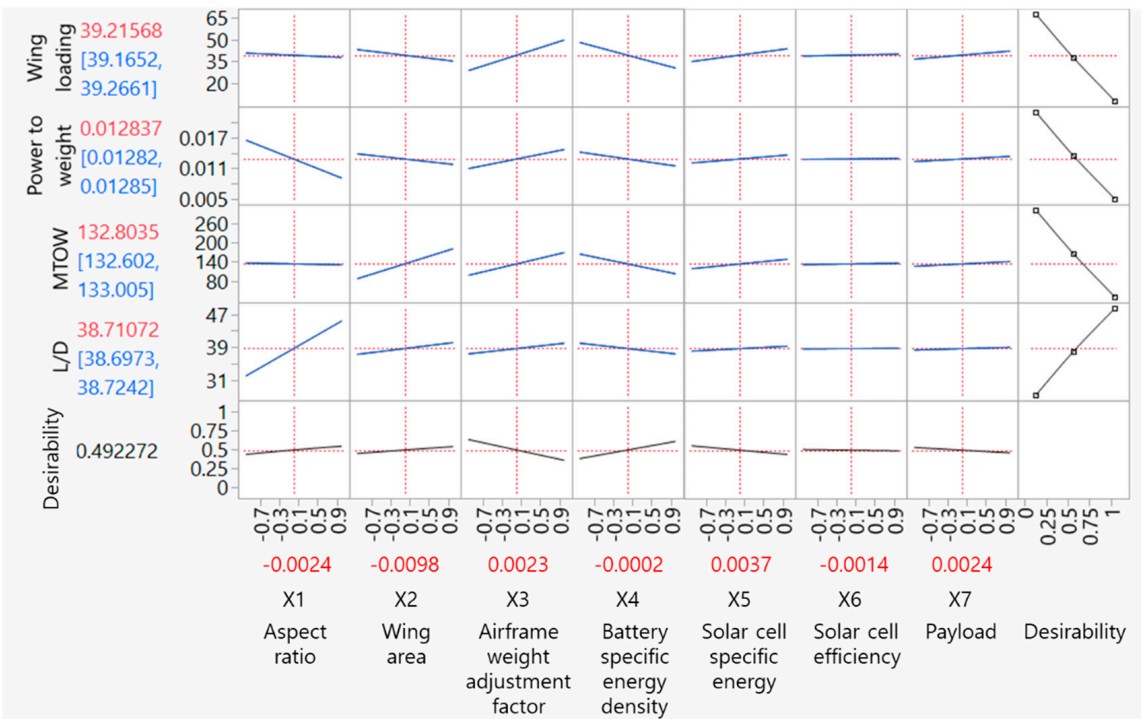

**Figure 19.** Desirability profile for independent design variables.

**Table 13.** Optimized independent design variables.

| Independent Design Variable | Desirability | Actual Value | Unit |
|---|---|---|---|
| Aspect ratio | −0.0024 | 19.98 | − |
| Wing area | −0.0098 | 34.85 | m$^2$ |
| Airframe weight adjustment factor | 0.0023 | 0.85 | − |
| Battery specific energy density | −0.0002 | 200 | Wh/kg |
| Solar cell specific mass | 0.0037 | 0.45 | kg/m$^2$ |
| Solar cell efficiency | −0.0014 | 0.195 | − |
| Payload weight | 0.0024 | 7.51 | kg |

**Table 14.** Optimized design objective parameters and constraints.

| Design Objective Parameter | Constraint | Actual Value | Unit |
|---|---|---|---|
| Wing loading | ≤30 | 38.38 | N/m$^2$ |
| Power to weight | ≤0.04 | 0.01 | hp/kg |
| Maximum take-off weight | ≤200 | 131.5 | kg |
| Lift to drag ratio | ≥35 | 39.5 | − |

*4.6. Performance Evaluation*

4.6.1. Performance of the Reference Aircraft

Figure 20 shows the solar power profile for 24 h according to the previously set reference aircraft. During the summer season (6/21), the solar power that can be obtained according to the main wing area of the aircraft is the largest, and the least on the winter solstice (12/22).

The total solar power that can be obtained during each season is shown in Figure 21a. The total solar power was also the largest in the summer and the smallest value in the winter solstice. Accordingly, as shown in Figure 21b, the energy margin in the summer solstice was 17.27% that is the only possible flight seasons.

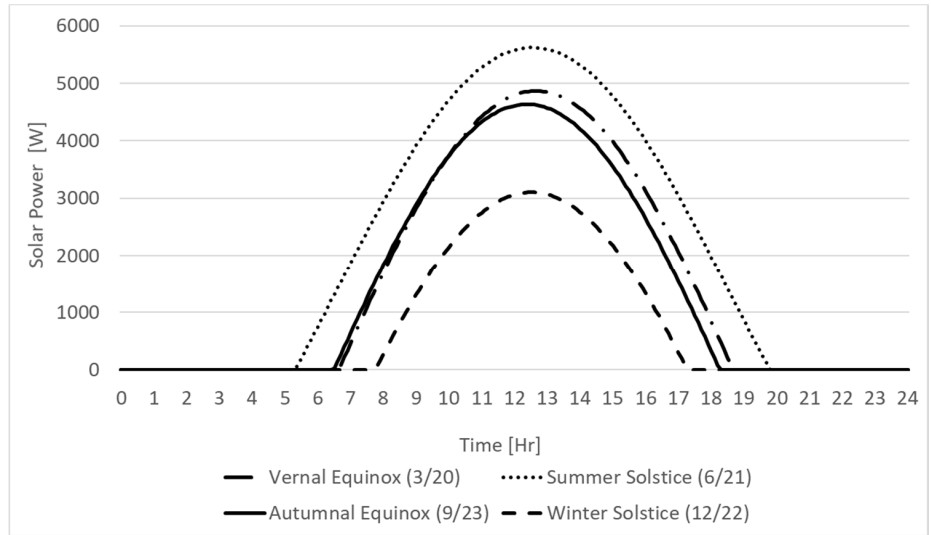

**Figure 20.** Solar power profile for the reference aircraft.

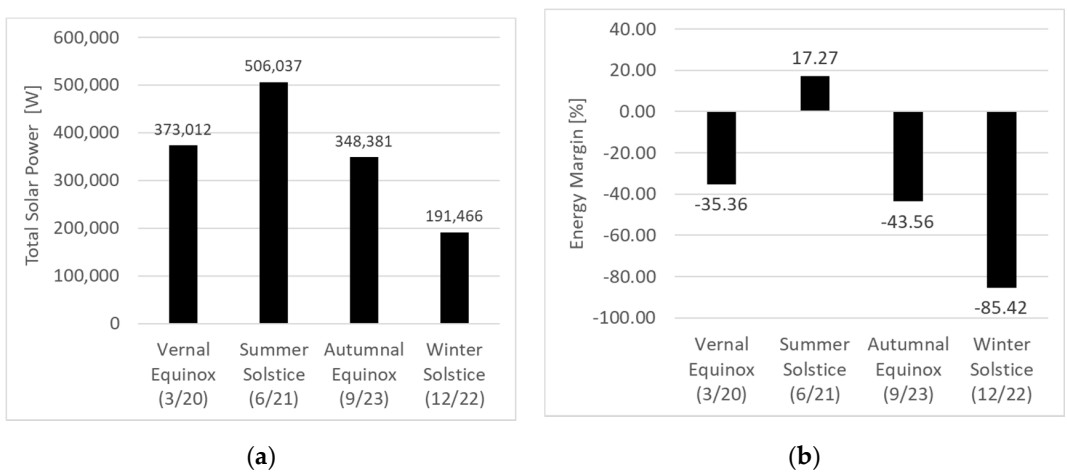

(**a**)                                                                                      (**b**)

**Figure 21.** (**a**) Total solar power for the reference aircraft; (**b**) energy margin for the reference aircraft.

We also analyzed how many days long-endurance is possible during the summer (6/21) as shown in Figure 22. The energy margin over 10% was assumed and on 5/24 and 7/19, the energy margin was 9.9%. Therefore, it was shown that the reference aircraft could fly for a total of 56 days, from 5/24 to 7/18.

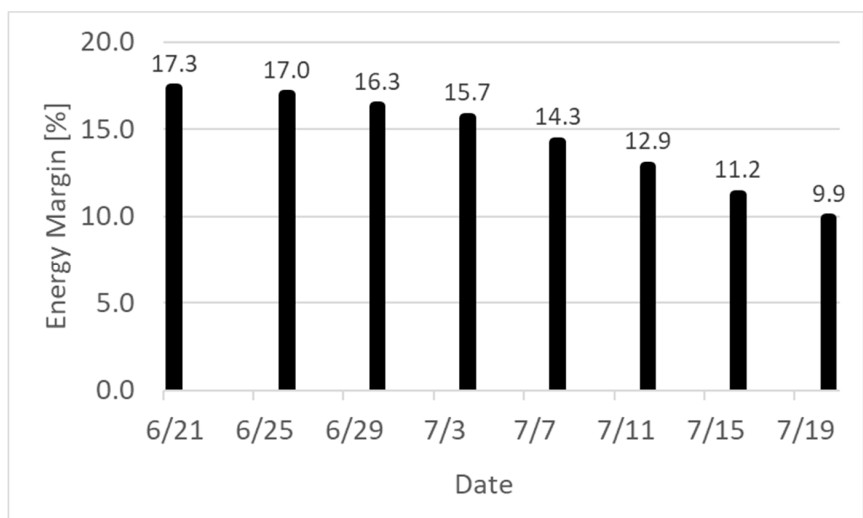

**Figure 22.** Long endurance performance analysis according to the energy margin for the reference aircraft.

### 4.6.2. Performance of the Optimized Aircraft

The photovoltaic power profile for 24 h for the optimized aircraft is also shown in Figure 23. Same as the reference aircraft, the solar power that can be obtained is the largest in the summer season (6/21), and the smallest in the winter solstice (12/22). The maximum solar power of the optimized aircraft is about 700 W higher than that of the reference aircraft on the summer solstice (6/21).

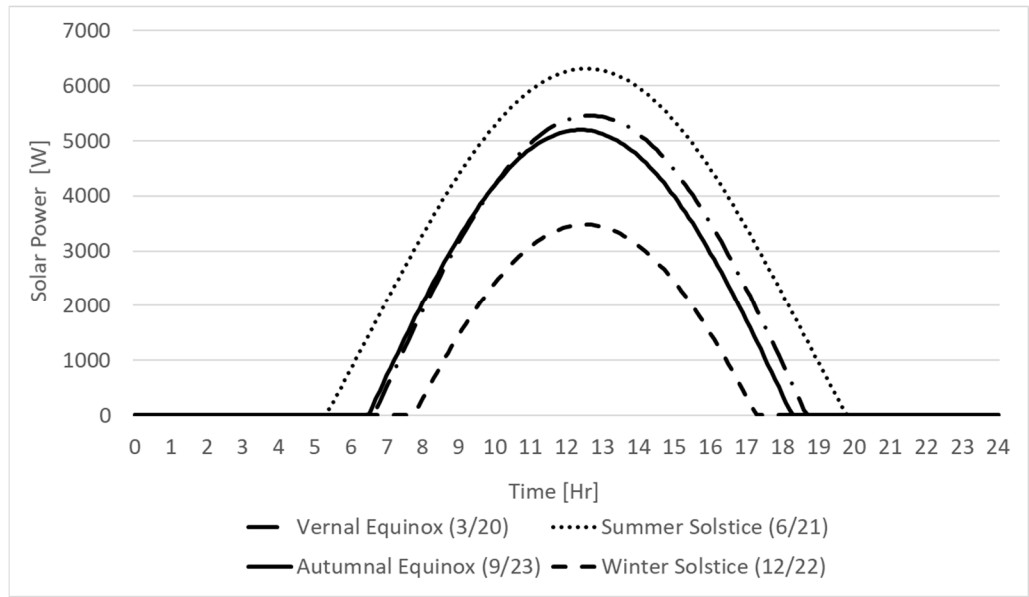

**Figure 23.** Solar power profile for the optimized aircraft.

The total solar power that can be obtained during each season is shown in Figure 24a. The total solar power was also the largest in the summer and the smallest in the winter solstice. As shown in Figure 24b, the energy margin in the summer solstice was 50.73%, showing that it is the only flight possible season.

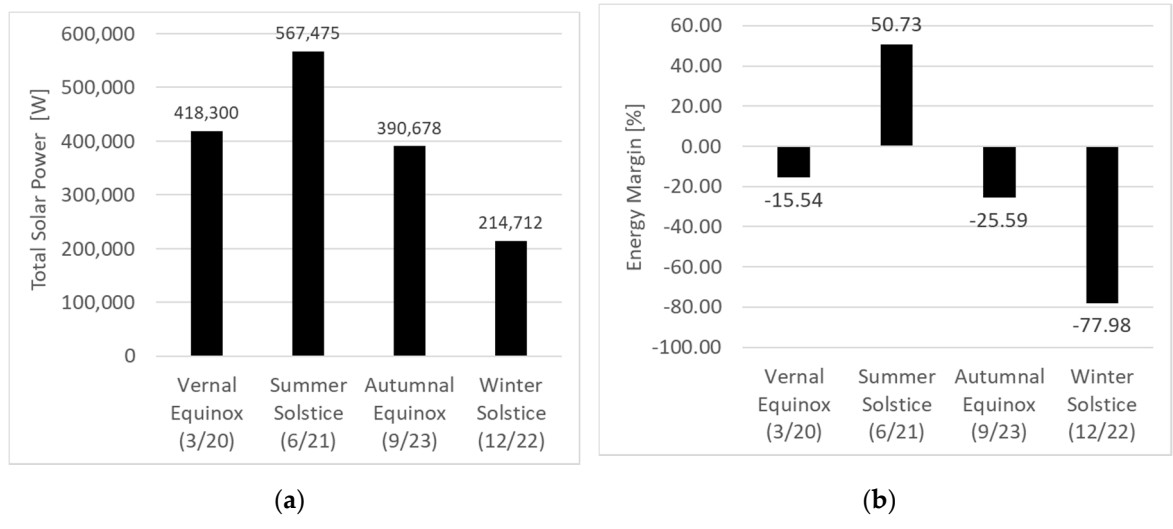

**Figure 24.** (**a**) Total solar power for the optimized aircraft; (**b**) energy margin for the optimized aircraft.

The duration of long endurance was estimated during the summer and is shown in Figure 25. The energy margin over 10% for safety was assumed and on 4/18 and 8/24, the energy margin was 9.1%. Therefore, it was shown that the optimized aircraft could fly for a total of 126 days, from 4/19 to 8/23.

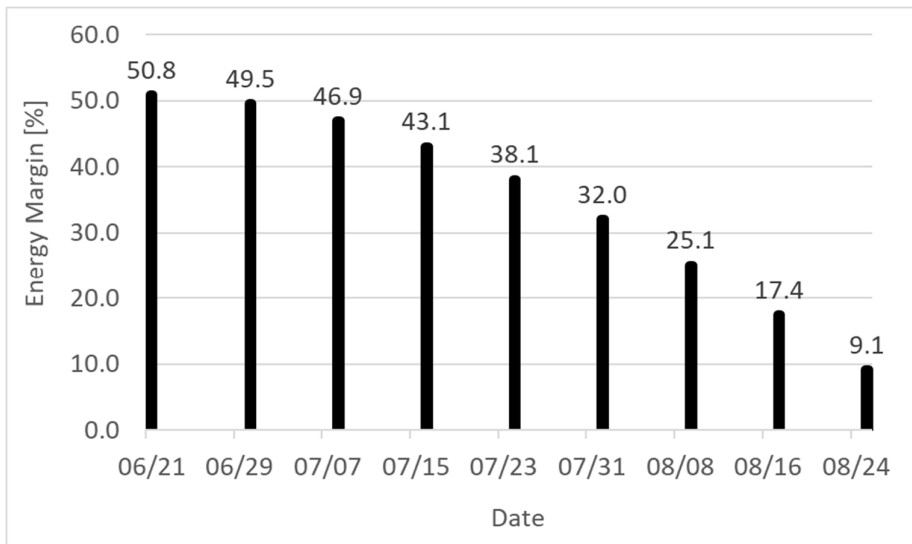

**Figure 25.** Long endurance performance analysis according to the energy margin for the optimized aircraft.

## 5. Conclusions

In this paper, a conceptual design and analysis of sensitivity of design parameters of HALE solar aircraft were conducted. Currently, research on HALE solar aircraft has been actively conducted, but most papers have determined the possibility of long-term flight or have studied initial sizing design. In this study, various detailed factors and subsystems that affect solar energy are considered, and the TIES method is introduced to analyze the sensitivity of the design variables. In addition, using the JMP program, the design variables are plotted to easily understand how they affect the performance of the aircraft.

In this study, a base configuration was established by referring to the configuration of Zephyr S, which is the most advanced solar aircraft so far in terms of long endurance performance. OpenVSP was employed to create configuration specifications such as wing area, wetted area, and aspect ratio. The aerodynamic characteristics of the base configuration were also found using OpenVSP. The solar aircraft design framework developed by our research team was used to implement modeling and simulation.

Since solar aircraft utilizing solar energy as their power source are largely affected by environmental factors the cruise altitude was set at 18 km in this study; place of flight was Anheung (latitude: $34.65^{\circ}$, longitude: $126.19^{\circ}$), Republic of Korea; flight date was the summer solstice (06/22). Under the conditions, required power and energy balance analysis was conducted using the solar aircraft design framework.

Through the design of the experiment, independent design variables were found that could affect design objective parameters; modeling of inter-factor relationships was performed. Using a screening test, seven variables were selected with the largest effects on the design objective parameters. Then, a regression equation was set up that expressed the relationship between the seven selected independent design variables and the design objective parameters. Through the data distribution ratio, the achieved regression equation was assessed and found to have sufficient reliability. After this, for more design variable combinations, 10,000 random numbers were generated to implement the Monte Carlo simulation.

Based on the results of the response surface method, sensitivity was analyzed under the conditions of the design objective parameters, with the horizontal axis representing the independent design variables and the vertical axis the performance. Aspect ratio (X1) was found to have a huge effect on factors of power to weight, and lift to drag ratio. Wing area (X2) was found greatly to affect the maximum take-off weight. This means that, if the aspect ratio is increased to increase the lift to drag ratio, these could be an adverse effect of reduced power to weight. Therefore, it is important to find a desired design point while mainly changing the independent design variables that are sensitive to the design requirements.

Based on the Monte Carlo simulation results, design feasibility of the design objective parameters was analyzed using the cumulative distribution function. The design probability needed to meet the wing loading was 20%; power to weight was 100%; maximum take-off weight was 92.4%; and lift to drag ratio was 75%. The wing loading has the lowest design probability; but still, it was possible to meet the design objective parameters within the range of the independent design variables set up previously.

Optimization was performed using desirability function, and optimized design independent values were derived corresponding to the desirability values of the selected independent design variables. Using HALE solar aircraft framework, the endurance performance was evaluated. In the case of Zephyr S, which is the reference configuration, it is possible to fly for about 56 days, from 5/24 to 7/18. In the case of the optimized aircraft, it is possible to fly for about 126 days, from 4/19 to 8/23. Therefore, the long endurance was improved by the methodology proposed in this study.

The variety of design parameters must be considered during the conceptual design phase of HALE solar aircraft. Since the main power source of solar aircraft is solar energy, there are various design variables that are not considered in the general aviation (GA) category aircraft design. For example, selection of a flight location for different latitudes and longitudes greatly affects solar energy output. Also, solar hour angle, earth declination angle, solar radiation, solar attenuation factor affect the solar aircraft performance. Therefore, it is necessary to predict and analyze the sensitivity of sizing and design feasibility of a HALE solar aircraft for changes in various combinations of design variables. There are many studies analyzing the possibility of long endurance flight that do not consider those various detailed design variable combinations. Since the power source is solar energy, the possibility of long endurance varies greatly depending on the detailed design variables, as described above, so this study introduced the design of experiment (DOE) to obtain the maximum information from the minimum number of experiments. Fractional factorial design, central composite design, which is a commonly used in the DOE method, was used. Also, the JMP program was used for accurate calculations. After analyzing the design feasibility, the optimization was performed using the desirability function of JMP software, and constraints were applied to each design objective parameter to derive the optimum values of independent design variables. Then, the values of optimized design independent variables were inputted to the solar aircraft design framework. Using this statistical method, the design possibility of long endurance performance can be identified in advance, thereby the time and cost for the conceptual design of HALE solar aircraft can be saved. By using the sensitivity analysis results of design variables, an optimized HALE solar aircraft can be designed that can improve endurance. Also endurance for the specific latitude and longitude can be predicted.

**Author Contributions:** J.-Y.Y., was responsible for all tasks related to the work, from establishing the methodology, to the entire data analysis; H.-Y.H., was responsible for conceptualization, the writing of the original manuscript, and managed the overall project and reviewed it. All authors have read and agreed to the published version of the manuscript.

**Funding:** This research was supported by the Research Grant from Sejong University through the Korea Agency for Infrastructure Technology Advancement funded by the Ministry of Land, Infrastructure and Transport of the Korean government (Project No.: 20CTAP-C157731-01).

**Conflicts of Interest:** The authors declare no conflict of interest.

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
