# Peer review of "Technology Identification, Evaluation, Selection, and Optimization of a HALE Solar Aircraft"

_applsci, doi:10.3390/app10217593_

Round 1
Reviewer 1 Report
Designing of flying objects is a complex and time-consuming process. This is due to the complexity of the used models and the multi-criteria nature of the design tasks being solved. In the reviewed paper, the authors presented the methodology of designing a very specific aircraft - HALE Solar Aircraft. The authors suggested the use of various tools and engineering methods in the design of HALE. Configuration selection and aerodynamic analyses were made with the use of OpenVSP and XFLR5 tools. The conceptual stage was carried out using the TIES methodology. The set of decision parameters was selected based on the analysis of the sensitivity of the solution to the change of the values of all design parameters. The relationship between the design (decision) parameters and the objective function was determined using the DOE methodology, which minimizes the number of analyses performed. Using the response surface method, regression equations and sensitivity profiles for the design variables were obtained. The possibility of achieving the assumed ranges of the variability of design parameters was made using the Monte Carlo simulation, for 10,000 cases. Design optimization was performed using the JMP tool to determine the optimal values of selected, most important design variables. The performance of the designed HALE Solar Aircraft, obtained with the use of computational methods, were compared with the performance of a real HALE, Zephyr S.
The choice of the HALE aircraft is very interesting because they are vehicles with high utility potential, but difficult to analyse due to the conditions of use and technological barriers. The proposed methodology is very interesting and allows to reduce the number of design analyses and design variables to the necessary minimum to obtain the optimal form of the aircraft. However, there are some questions and doubts that need to be clarified by the authors:
- is the use of Low Fidelity methods (XFLR5) not cause too large modelling errors and thus incorrect conclusions about the sensitivity of the objective function to the change of design variables?
concerning the above
- Is the applied aerodynamic model too simplified for this type of analysis? It would be more appropriate to use multi-fidelity methods.
- The weight model is excessively simplified, which does not allow to find some dependencies that certainly exist (e.g. between Battery efficiency and HALE weight, which determines the wing loading, power loading and so on).
- The Airframe weight adjustment factor needs clarification. What exactly does it mean? Why is it a design variable if it results from the weight of the structure, which is a calculated quantity?
- Why are design variables called independent, although there are relationships between them that affect the boundary values (target values)?
- Why is the Payload weight design variable, if it is a required value resulting from the weight of the onboard equipment?
- How was the performance of the Zephyr S obtained and can it be compared with the designed HALE Solar Aircraft, since a very simplified model was used at certain stages of the design?
- Was the approach (methodology) validated?
- does the conclusion regarding e.g. target value of wing loading be reached with a probability of 20% does not mean that the initial target value has been set incorrectly and should be corrected?
However, the questions and comments formulated do not change my high assessment of the presented work.
Reviewer 2 Report
This manuscript presents a sensitivity analysis and optimization of High Altitude Long Endurance (HALE) solar aircraft employing the TIES framework. This framework provides new insight about design parameters that can enhance the parameter of HALE solar aircraft. As it stands, the manuscript is fairly complete and only requires minor changes. Below are suggestions for the authors to consider.
- Units of distance and area need to have consistent font styles in Tables 1, 3, 6, 8 and 10.
- For thoroughness, please describe in the text what the symbols rho and V represent.
- It is unclear whether the attenuation factors presented in Figure 4 were derived from experiments conducted by the authors themselves or if these values were obtained from previous work. If the latter case, please include appropriate references.
- Equations (3) or (4) do not include a K coefficient. Therefore, the text in line 137 needs to be modified to reflect the K_1 and K_2 coefficients in Equations (4) accurately.
